# Subthalamic nucleus stabilizes movements by reducing neural spike variability in monkey basal ganglia

Taku Hasegawa [1], Satomi Chiken [1,2], Kenta Kobayashi [2,3] & Atsushi Nambu [1,2,3✉]

The subthalamic nucleus projects to the external and internal pallidum, the modulatory and output nuclei of the basal ganglia, respectively, and plays an indispensable role in controlling voluntary movements. However, the precise mechanism by which the subthalamic nucleus controls pallidal activity and movements remains elusive. Here, we utilize chemogenetics to reversibly reduce neural activity of the motor subregion of the subthalamic nucleus in three macaque monkeys (*Macaca fuscata*, both sexes) during a reaching task. Systemic administration of chemogenetic ligands prolongs movement time and increases spike train variability in the pallidum, but only slightly affects firing rate modulations. Across-trial analyses reveal that the irregular discharges in the pallidum coincides with prolonged movement time. Reduction of subthalamic activity also induces excessive abnormal movements in the contralateral forelimb, which are preceded by subthalamic and pallidal phasic activity changes. Our results suggest that the subthalamic nucleus stabilizes pallidal spike trains and achieves stable movements.

[1] Division of System Neurophysiology, National Institute for Physiological Sciences, Okazaki, Aichi 444-8585, Japan. [2] Department of Physiological Sciences, SOKENDAI, Okazaki, Aichi 444-8585, Japan. [3] Section of Viral Vector Development, National Institute for Physiological Sciences, Okazaki, Aichi 444-8585, Japan. ✉email: nambu@nips.ac.jp

The subthalamic nucleus (STN) is a small nucleus, but occupies an important position in the basal ganglia (BG) circuitry. The STN receives cortical inputs directly through the cortico-STN pathway and indirectly through the cortico-striato-external pallido (GPe)-STN pathway[1] and sends a glutamatergic projection to the GPe and internal pallidum (GPi). The GPe innervates all other nuclei in the BG[2,3], whereas the GPi is the output nucleus of the BG[4,5]. Therefore, the STN affects the activity of all BG nuclei as well as the output nucleus.

The STN also plays pivotal roles in normal functions and disease conditions of the BG. Lesions or chemical blockade of the STN induces involuntary movements known as hemiballism (Supplementary Table 1)[6–8]. Abnormal activity of STN neurons has been reported in Parkinson's disease (PD)[9–12], and it is suggested that the reciprocal excitatory and inhibitory connection between the STN and GPe is the source of pathologic oscillation associated with PD[13,14]. Moreover, lesions or deep brain stimulation (DBS) in the STN can ameliorate the motor symptoms of PD[15–17].

These clinical effects are consistent with the classical BG model; the cortico-striato-GPi *direct* pathway facilitates movements, whereas the cortico-STN-GPi *hyperdirect* and cortico-striato-GPe-STN-GPi *indirect* pathways suppress movements[5,8,18,19]. This model hypothesizes that the STN inhibits and/or cancels movements, which is supported by human neuroimaging and electrophysiologic recording/stimulation studies of the STN[20–22]. However, the activity of STN neurons changes in relation to simple limb or eye movements as well[23,24], and such movement-related activity is not easily explained from the perspective of movement suppression. It has been argued that the STN activates antagonist muscles necessary for stopping movements or suppresses competing motor programs, thus allowing the *direct* pathway to release only a selected motor program[18,19,25]. Furthermore, pharmacologic activation of the STN induces involuntary movements on the contralateral side[26,27]. These previous observations suggest that the STN endows the BG circuitry with more complex neural computations than the simple dichotomy of movement facilitation and suppression.

To clarify the functional role of the STN in motor control, in the present study, we utilized the Designer Receptors Exclusively Activated by a Designer Drug (DREADD) technology to manipulate the neural activity in the STN of macaque monkeys. Although DREADDs have been utilized widely in rodents, application in sub-human primates is rather limited and only one electrophysiologic evaluation at the single-neuron level[28] is reported. Here, we showed that administration of DREADD ligands moderately reduced STN activity, which was sufficient to induce abnormal involuntary movements and extend the movement time. Single-unit recordings revealed that pauses and spike train variability increased in both GPe and GPi neurons, whereas their movement-related activity was slightly affected. Our findings thus suggest a role for the STN in stabilizing spike trains of GPe/GPi neurons to achieve stable motor control.

## Results

**Moderate reduction of STN activity using DREADD.** The STN motor subregion involved in control of the forelimbs was identified based on neuronal responses to electric stimulation of the forelimb regions of the primary motor cortex (M1) and supplementary motor area (SMA), i.e., biphasic early and late excitation via the cortico-STN *hyperdirect* and cortico-striato-GPe-STN *indirect* pathways, respectively (Fig. 1a, magenta and green arrows)[1,8,19,29]. An adeno-associated virus (AAV) vector that co-expresses an inhibitory DREADD receptor, hM4Di, and enhanced green fluorescent protein (EGFP) was injected

unilaterally into the identified STN motor subregion. Histologic examination of monkeys E and K revealed that EGFP expression was found in the dorsolateral part of the posterior STN (Fig. 1b), corresponding to the motor subregion[1,8,24,29–32], with preference to neurons (Supplementary Fig. 1a, b) and similar transduction efficiencies in both monkeys (60.2% and 68.3% of neurons in the dorsolateral part of the posterior STN in monkeys E and K, respectively; Supplementary Fig. 1c). However, a small fraction of transduced cells were found dorsal to the STN such as the thalamus and zona incerta, presumably due to the leakage of AAV solution along the injection track (Supplementary Fig. 2a–c, g). Transduced cells in the thalamus avoided its forelimb region (Supplementary Fig. 2h) and would have negligible effects on forelimb movements. No sign of retrograde transduction was found in the GPe/GPi (Supplementary Fig. 2d–f, g).

After receptor expression (>3 weeks), neuronal recordings and behavioral observations were initiated. A ligand for hM4Di, clozapine N-oxide (CNO; 1 mg/kg, i.v., to monkey E) or newly developed deschloroclozapine (DCZ; 0.1 mg/kg, i.m., to monkeys K and U)[33], was administered systemically. The efficacy of chemogenetic suppression was determined by recording unit activity in the STN. The baseline firing rates of the STN neurons began to decrease at 15 min after CNO administration and 5 min after DCZ administration (Fig. 1c, d), consistent with the rapid delivery of DCZ to the brain[33]. The firing rates decreased to 65-75% of the rates before ligand administration (monkey E, 73.6 ± 8.4%, $P < 0.001$, $n = 12$; monkey K, 67.0 ± 5.0%, $P < 0.001$, $n = 20$; monkey U, 66.0 ± 6.3%, $P < 0.001$, $n = 17$; two-tailed Wilcoxon-signed rank test), whereas administration of vehicle (VEH) had no effect (monkey K, 91.7 ± 5.8%, $P = 0.1$, $n = 15$).

Lesions or chemical blockade of the STN induces involuntary movements known as hemiballism[6–8]. In the present study, administration of DREADD ligands often induced involuntary movements in all three monkeys, i.e., choreic/ballistic movements of the shoulder, elbow, wrist, and fingers contralateral to the AAV injection side while the monkeys sat quietly in a chair without performing any task (Supplementary Movie 1; Fig. 1e). No abnormal movements were noted with other body parts, such as the eyes, hindlimbs, or ipsilateral forelimb. The involuntary movements began approximately 20 min (CNO) or 5 min (DCZ) after ligand administration and continued for >2 h. Involuntary movements following ligand administration were observed repeatedly throughout the experimental periods (58, 85, and 75 weeks after AAV vector injection in monkeys E, K, and U, respectively). These histologic, electrophysiologic, and behavioral observations indicated that the forelimb motor subregion of the STN was successfully targeted and that DREADD receptor expression was stable for >1 year.

**Movements disturbed by reduction of STN activity.** Each monkey was trained to perform a reach-and-pull task composed of externally triggered (ET) and self-initiated (SI) trials using its contralateral hand (Fig. 2a); the monkey was required to initiate reaching either immediately after "Go" cue presentation in ET trials or with a delay of 1.5 s without any explicit Go cue in SI trials. ET and SI trials were randomly presented, and the trial type was indicated by the color of an LED (Task cue): blue for ET trials and red for SI trials. In ET trials, the blue LED turned green (Go cue) with a delay of 1–2 s, and then the monkey must release the home lever within 0.5 s and pull the front lever to be rewarded. In SI trials, the monkey must wait and maintain the home position for >1.5 s after Task cue and then initiate reaching movements.

After CNO or DCZ administration, reaching motions became unstable, and the monkeys required more time to grab the front

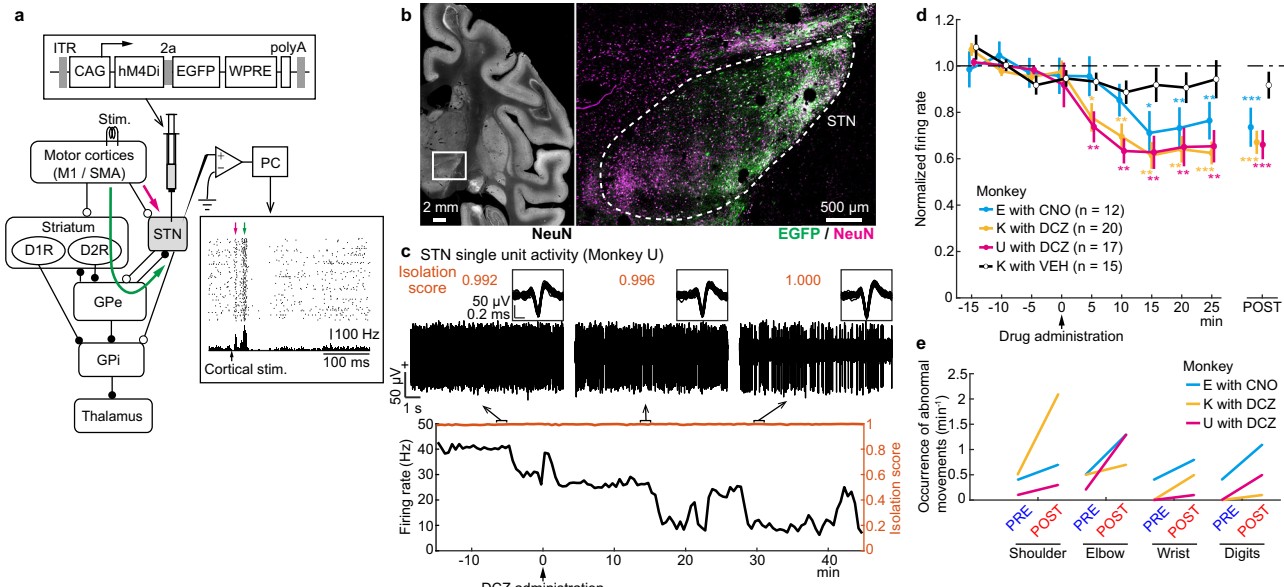

**Fig. 1 Reduction of STN activity using DREADD. a** The motor subregion of the STN was identified by the characteristic biphasic excitation (bottom right inset) induced by motor cortical stimulation, followed by infusion of an AAV vector co-expressing hM4Di and EGFP (top inset). Magenta and green arrows indicate early and late excitation. **b** Histologic confirmation of AAV transduction with anti-GFP (green) and NeuN (magenta) antibodies in monkey K. The brain region indicated by an open box on the left is enlarged on the right. The immunostaining was repeated twice with similar results. **c** Firing rate changes (bottom, black) and Isolation scores (brown) of a STN neuron with DCZ administration. Overlay of 50 isolated spikes (inset) and raw extracellular signals (top) at indicated time points are also shown. **d** Time course of effects of CNO (1.0 mg/kg, i.v.), DCZ (0.1 mg/kg, i.m.), or vehicle (VEH) on baseline firing rates of STN neurons. STN activity was normalized based on activity during the PRE period (from −15 to 0 min) and averaged in 5-min bins (left) and in the POST periods (right; from 15 to 25 min after CNO administration; from 10 to 25 min after DCZ/VEH administration). Firing rates significantly decreased at 15, 20, and 25 min ($P = 0.01$, 0.004, and 0.004, respectively; one-tailed Wilcoxon-signed rank test with Bonferroni correction) and in the POST (0.0005; two-tailed Wilcoxon-signed rank test) after CNO in monkey E, at 5, 10, 15, 20, and 25 min (0.02, 0.001, 0.001, 0.001, and 0.0007) and in the POST (0.0002) after DCZ in monkey K, and at 5, 10, 15, 20, and 25 min (0.005, 0.002, 0.003, 0.009, and 0.003) and in the POST (0.0004) after DCZ in monkey U. Error bars indicate SEM. *$P < 0.05$, **$P < 0.01$, ***$P < 0.001$. **e** Occurrence of abnormal movements at rest during the PRE and POST (from 20 to 60 min after CNO/DCZ administration) periods, quantified from a single session in each animal.

lever in some ET and SI trials, as observed by an increase in movement time (MT) in all three monkeys (Fig. 2b *right*; $P < 0.01$ for all monkeys in both ET and SI trials; two-tailed Wilcoxon-signed rank test; $n = 14$, 15, and 20 sessions for monkeys E, K, and U, respectively). The success rate (Fig. 2b *left*; monkey E, $P < 0.05$; monkey U, $P < 0.01$) and reaction time (RT; Fig. 2b *middle*; monkey E, $P < 0.05$; monkey U, $P < 0.001$) of monkeys E and U decreased in SI trials. There was no change in the success rate and RT in the ET trials, except for a decrease in RT of monkey U (Fig. 2b *middle*; $P < 0.05$). No significant effects were observed on the success rate, RT, or MT after VEH administration (Supplementary Fig. 3a; $P > 0.05$ for all monkeys in both ET and SI trials; two-tailed Wilcoxon-signed rank test; $n = 8$, 12, and 10 for monkeys E, K, and U, respectively), or in task performance using the ipsilateral hand (Supplementary Fig. 3b; $P > 0.05$ for all monkeys in both ET and SI trials; $n = 8$, 9, and 9), indicating that any off-target effects of the DREADD ligands or their effects on STN non-motor functions were minimal.

Three-dimensional (3D) trajectories of the contralateral shoulder, elbow, wrist, and hand during reaching movements were reconstructed from RGB (x–y) and depth (z) images in monkeys K and U (Supplementary Fig. 4), and trajectories of the wrist position were statistically analyzed (Fig. 2c). The trajectory deviation increased in both monkeys (Fig. 2c *left* Movement): monkey K (ET trials, $P < 10^{-39}$, $n = 179$; SI trials, $P < 10^{-28}$, $n = 155$; two-tailed Mann–Whitney $U$-test) and monkey U (ET, $P < 10^{-14}$, $n = 101$; SI, $P < 10^{-8}$, $n = 103$). The maximum speed increased in both monkeys (Fig. 2c *middle* Movement): monkey K (ET, $P < 0.01$; SI, $P < 0.01$) and monkey U (ET, $P < 10^{-8}$; SI,

$P < 10^{-5}$). The trajectory tortuosity, a measure of the bending and curving of a path, increased in monkey K (Fig. 2c *right*; ET, $P < 10^{-16}$; SI, $P < 10^{-9}$) but not monkey U (ET, $P = 0.6$; SI, $P = 0.5$). These results indicate that the longer MT in the POST period (Fig. 2b *right*) was due to high variability in reaching movements, rather than slow movements. In addition, analysis of the trajectory during the delay period (i.e., at rest) revealed task-irrelevant involuntary movements. The trajectory deviation increased in both monkeys (Fig. 2c *left* Rest): monkey K (ET, $P < 10^{-47}$; SI, $P < 10^{-40}$) and monkey U (ET, $P < 10^{-12}$; SI, $P < 10^{-13}$). The maximum speed increased in both monkeys (Fig. 2c *middle* Rest): monkey K (ET, $P < 10^{-7}$; SI, $P < 10^{-5}$) and monkey U (ET, $P < 10^{-4}$; SI, $P < 0.05$). These changes were not observed in either monkey after VEH administration: monkey K (Supplementary Fig. 3c; $P > 0.05$ for all measures and conditions; $n = 74$ ET and 70 SI trials) and monkey U ($P > 0.05$; $n = 45$ ET and 54 SI trials). Thus, excessive movements occurred both during movements and at rest, resulting in disturbed reaching and involuntary movements, respectively.

Involuntary movements were also detected as positional fluctuations of the home lever pulled by monkeys at rest. In monkeys K and U, DCZ administration induced involuntary movements more frequently than VEH administration before and after Task cue in both ET and SI trials (Fig. 2d): before Task cue (Fig. 2e; monkey K, 6.4% and 15.3% for VEH [$n = 31$] and DCZ [$n = 43$], $P < 0.001$; monkey U, 10.7% and 21.6% for VEH [$n = 18$] and DCZ [$n = 41$], $P < 0.001$; two-tailed Mann–Whitney $U$-test), after Task cue in ET trials (monkey K, 14.1% and 25.3%, $P < 0.01$; monkey U, 13.0% and 26.7%, $P < 0.01$), and after Task

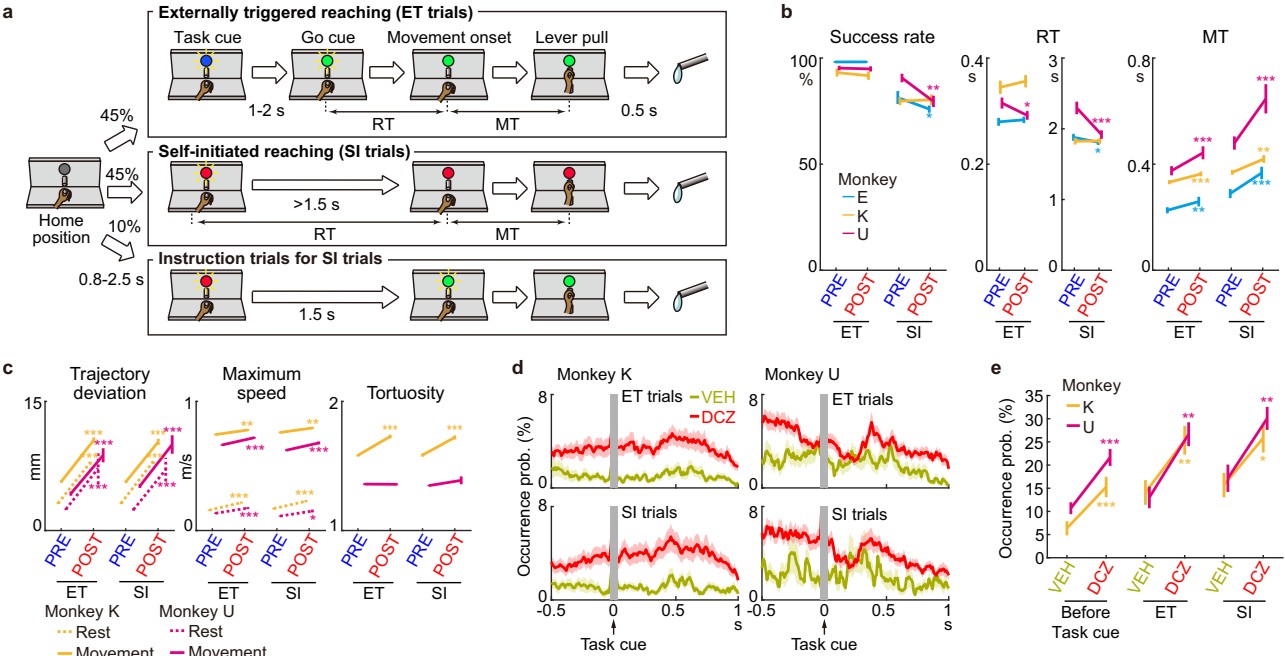

**Fig. 2 Motor effects of reduction of STN activity on performance of a reach-and-pull task. a** Custom reach-and-pull task. RT, reaction time; MT, movement time. **b** Task performance using the hand contralateral to the AAV injection side before (PRE) and after (POST) DREADD ligand administration (14, 15, and 20 sessions in monkeys E, K, and U, respectively). Success rate decreased in SI trials ($P = 0.02$ and 0.001, monkeys E and U, respectively; two-tailed Wilcoxon-signed rank test), RT decreased in ET (0.01, U) and SI (0.02 and 0.0003, E and U), and MT increased in ET (0.007, $6 \times 10^{-5}$, and 0.0007; E, K, and U) and SI (0.0001, 0.001, and 0.0002; E, K, and U). **c** Analyses of wrist trajectories (179 ET and 155 SI trials in monkey K, and 101 ET and 103 SI in U). The following parameters significantly increased: Trajectory deviation during Rest (ET trials, $P = 4 \times 10^{-48}$ and $1 \times 10^{-13}$, monkeys K and U, respectively; SI trials, $2 \times 10^{-41}$ and $4 \times 10^{-14}$; two-tailed Mann–Whitney $U$-test) and Movement (ET, $4 \times 10^{-40}$ and $1 \times 10^{-15}$; SI, $2 \times 10^{-29}$ and $2 \times 10^{-9}$), Maximum speed during Rest (ET, $4 \times 10^{-8}$ and $3 \times 10^{-5}$; SI, $3 \times 10^{-6}$ and 0.02) and Movement (ET, 0.003 and $1 \times 10^{-9}$; SI, 0.003 and $1 \times 10^{-6}$), and Tortuosity (ET, $1 \times 10^{-16}$, K; SI, $4 \times 10^{-10}$, K). **d**, **e** Occurrence of involuntary movements during the POST period with the VEH (31 and 18 sessions in monkeys K and U) or DCZ (43 and 41) administration. Solid lines and shading indicate mean ± SEM with 1-ms bin (**d**). Statistical analysis during the 0.5 s preceding (Before Task cue) and 1 s following Task cue (ET and SI) between VEH and DCZ (**e**). Involuntary movements increased during Before Task cue ($P = 0.0006$ and 0.0002, monkeys K and U; two-tailed Mann–Whitney $U$-test), ET (0.007 and 0.006), and SI (0.03 and 0.003). Error bars indicate SEM. *$P < 0.05$, **$P < 0.01$, ***$P < 0.001$.

cue in SI trials (monkey K, 15.7% and 25.6%, $P < 0.05$; monkey U, 17.2% and 30.1%, $P < 0.01$).

Electromyogram (EMG) of the biceps brachii and triceps brachii muscles of monkeys K and U was obtained during the task (Supplementary Fig. 5a1, a2). Task-irrelevant phasic EMG activity was observed, corresponding to involuntary movements. In the POST period, abnormal EMG activity was induced more frequently after Task cue (Supplementary Fig. 5b1, b2), and the movement-related EMG activity tended to increase (Supplementary Fig. 5a1, a2), consistent with the abovementioned trajectory analyses.

**Diminished cortically evoked responses in GPe/GPi neurons.** Single-unit GPe/GPi activity was recorded using a 16-channel linear electrode in 78 sessions. Of 192 neurons (100 GPe and 92 GPi) examined, 168 (82 GPe and 86 GPi) responded to M1 and/ or SMA stimulation and were classified as "high-frequency discharge and pause" (HFD-P) GPe, "low-frequency discharge and burst" (LFD-B) GPe, "high-frequency discharge" (HFD) GPi, or "low-frequency discharge" GPi neurons based on firing rates and patterns (Table 1)[34]. HFD-P GPe neurons (78/82) and HFD GPi neurons (83/86) were major and further analyzed, whereas LFD-B GPe neurons (4/82) are shown separately (Supplementary Fig. 6). STN activity was also recorded in another 32 sessions. Of 52 STN neurons examined, 37 responded to M1 and/or SMA stimulation (Table 1).

To examine how reduction of STN activity alters cortical inputs to the GPe/GPi through the STN, neuronal responses to

cortical stimulation in the STN and GPe/GPi were recorded (Supplementary Fig. 7). During the PRE period, the typical response of STN neurons was biphasic, consisting of early and late excitation phases, conveyed through the cortico-STN and cortico-striato-GPe-STN pathways, respectively[1,8], while that of GPe/GPi neurons was triphasic, consisting of early excitation, inhibition, and late excitation phases, conveyed through the cortico-STN-GPe/GPi, cortico-striato-GPe/GPi, and cortico-striato-GPe-STN-GPe/GPi pathways, respectively[8,35]. During the POST period, spontaneous firing rate in the STN was decreased (Fig. 1d); however, cortically induced early and late excitation remined unchanged (Supplementary Fig 7a–c, *top*). In the GPe/GPi, early excitation was diminished, and inhibition was enhanced (Supplementary Fig. 7a–c *middle* and *bottom*), suggesting that excitatory inputs from the STN to the GPe/GPi were reduced and that inhibitory inputs from the striatum were relatively enhanced. Late excitation was increased in the GPe, which possibly reflected preceding enhanced inhibition through the inhibitory feedback GPe-STN-GPe pathway. Baseline activity decreased significantly in the GPe but not GPi. Hence, reduction of STN activity decreased the efficiency of information transmission from the cortex to the GPe/GPi via the STN, as well as the baseline activity of the GPe.

**GPe/GPi movement-related activity was weakly affected after CNO/DCZ.** Since movement-related activity in the GPe/GPi was qualitatively similar between the ET and SI trials as shown in

**Table 1 Number of STN, GPe, and GPi neurons examined during task performance.**

| | Firing patterns | Monkey | | | Total | Firing rate |
|---|---|---|---|---|---|---|
| | | E | K | U | | (mean ± SD, Hz) |
| GPe | HFD-P | - | 39 (35/4/0) | 39 (27/12/0) | 78 (62/16/0) | 76.5 ± 29.5 |
| | LFD-B | - | 4 (3/1/0) | 0 | 4 (3/1/0) | 10.7 ± 7.8 |
| | Total | | | | 82 | |
| GPi | HFD | 20 (20/0/0) | 33 (22/11/0) | 30 (19/10/1) | 83 (61/21/1) | 80.5 ± 31.5 |
| | Low-frequency discharge | 3 (3/0/0) | 0 | 0 | 3 (3/0/0) | 8.0 ± 3.0 |
| | Total | | | | 86 | |
| STN | | a | 20 (20/0/0) | 17 (15/1/1) | 37 (35/1/1) | 28.3 ± 16.8 |

Based on firing rates and patterns, GPe and GPi neurons were classified as "high-frequency discharge and pause" (HFD-P) GPe (mean firing rate ≥20 Hz), "low-frequency discharge and burst" (LFD-B) GPe (<20 Hz), "high-frequency discharge" (HFD) GPi neurons (≥20 Hz), or "low-frequency discharge" GPi neurons (<20 Hz). "Low-frequency discharge" GPi neurons could be cholinergic neurons surrounding the GPi. Numbers in each parenthesis indicate numbers of neurons responded to only M1, both M1 and SMA, and only SMA stimulation, respectively. Spontaneous firing rates were measured during the 500 ms before Task cue in the PRE period.
<sup>a</sup>Activity of 12 STN neurons was recorded without task performance to examine the effect of CNO administration in monkey E.

Supplementary Fig. 8, the results of ET trials are mainly presented in this section.

All GPe neurons with M1/SMA inputs exhibited movement-related activity during the PRE period and were thus classified as increasing (INC) type (43/78, 55%) or decreasing (DEC) type (35/78, 45%) based on the polarity of the largest movement-related modulation (see Methods for details). GPe neurons in each type are exemplified in Supplementary Fig. 9a, b. The activity of GPe neurons was summarized as heatmaps (Fig. 3a) and population peri-event time histograms (PETHs; Fig. 3b). In the POST period, the firing rate decreased in both types of GPe neurons, but the temporal structure of the movement-related modulation was preserved, i.e., INC- and DEC-type neurons showed increased and decreased activity, respectively, during movements. Statistical analyses revealed that the onset timing of movement-related modulation was similar for each neuron type between the PRE and POST periods (Fig. 3c3; INC type, from $153 \pm 35$ ms to $195 \pm 37$ ms, $P = 0.08$, $n = 43$; DEC type, from $107 \pm 31$ ms to $127 \pm 30$ ms, $P = 0.2$, two-tailed Wilcoxon-signed rank test; mean ± SEM). Also, the duration of movement-related modulation did not change (Fig. 3c4; INC, from $245 \pm 22$ ms to $254 \pm 26$ ms, $P = 0.8$; DEC, from $246 \pm 32$ ms to $219 \pm 20$ ms, $P = 0.7$). However, the following changes were observed in the POST period: (1) The baseline firing rate decreased (Fig. 3c1; INC, from $70.6 \pm 4.3$ Hz to $61.3 \pm 4.4$ Hz, $P < 0.01$; DEC, from $83.8 \pm 5.0$ Hz to $68.8 \pm 5.1$ Hz, $P < 10^{-4}$); (2) The peak and trough amplitudes of the PETHs decreased (Fig. 3c2; peak amplitude in INC, from $69.5 \pm 4.1$ Hz to $58.5 \pm 4.5$ Hz, $P < 0.01$; trough amplitude in DEC, from $60.3 \pm 3.9$ Hz to $52.0 \pm 4.3$ Hz, $P < 0.05$).

Similarly, all GPi neurons with M1/SMA inputs exhibited movement-related activity in the PRE period and were classified as either INC type (33/83, 40%) or DEC type (50/83, 60%). GPi neurons in each type are exemplified in Supplementary Fig. 9c, d. The activity of GPi neurons was also summarized as heatmaps (Fig. 3d) and population PETHs (Fig. 3e). Movement-related activity was affected only weakly by reduction of STN activity. Statistical analyses indicated that the following parameters did not change during the POST periods: (1) baseline firing rate (Fig. 3f1; INC, from $80.1 \pm 5.2$ Hz to $76.4 \pm 6.1$ Hz, $P = 0.2$, $n = 33$; DEC, from $85.1 \pm 4.0$ Hz to $85.2 \pm 3.6$ Hz, $P = 0.9$, $n = 50$); (2) PETH trough amplitude of DEC-type neurons (Fig. 3f2; from $61.5 \pm 3.4$ Hz to $59.1 \pm 3.7$ Hz, $P = 0.3$); (3) onset timing of movement-related modulations (Fig. 3f3; INC, from $91 \pm 40$ ms to $94 \pm 46$ ms, $P = 0.7$; DEC, from $125 \pm 35$ ms to $93 \pm 29$ ms, $P = 0.6$); and (4) duration of movement-related modulations (Fig. 3f4; INC, from $218 \pm 22$ ms to $264 \pm 28$ ms, $P = 0.1$; DEC, from $199 \pm 14$ ms to $192 \pm 17$ ms, $P = 0.3$).

PETH peak amplitude of INC-type neurons was exceptional (Fig. 3f2; from $74.1 \pm 5.7$ Hz to $65.4 \pm 5.8$ Hz, $P < 0.05$).

The same analyses were performed for STN neurons (Supplementary Fig. 10). All STN neurons with M1/SMA inputs exhibited movement-related activity in the PRE period and were classified as INC type (25/37, 68%) or DEC type (12/37, 32%). In the POST period, both types exhibited significantly reduced baseline firing rates, but movement-related activity was affected only weakly (Supplementary Fig. 10c–e). The above analyses with the PETH showed that the activity of GPe/GPi neurons was affected only weakly by reduction of STN activity. Trial-averaging analyses such as PETHs may not be appropriate to explain trial-to-trial variability in task performance. Hence, the temporal structures of spike trains (e.g., bursts and pauses) were analyzed in detail.

**STN/GPe/GPi spike train variability increased after CNO/DCZ.** Figure 4a shows examples of bursts and pauses in STN/GPe/GPi neurons. In the POST period, the pauses (blue lines) tended to become more frequent and of longer duration both at rest and during movements in all three examples, and burst activity (red lines) became more frequent in GPe neurons. Statistical analyses at rest and during movements revealed that the pause probability increased in the STN/GPe/GPi during the POST period (Fig. 4b; STN, from $20.0 \pm 3.3\%$ to $26.7 \pm 3.8\%$, $P < 0.01$, $n = 37$; GPe, from $19.7 \pm 1.7\%$ to $27.6 \pm 2.1\%$, $P < 10^{-7}$, $n = 78$; GPi, from $8.0 \pm 1.0\%$ to $10.5 \pm 1.3\%$, $P < 0.001$, $n = 83$; two-tailed Wilcoxon-signed rank test; mean ± SEM), whereas the burst probability increased only in GPe neurons (from $0.28 \pm 0.05\%$ to $0.55 \pm 0.09\%$, $P < 0.001$). The coefficient of variation of inter-spike intervals (CV of ISIs) increased in the STN/GPe/GPi during the POST period (Fig. 4b; STN, from $1.05 \pm 0.07$ to $1.27 \pm 0.07$, $P < 10^{-4}$; GPe, from $1.15 \pm 0.05$ to $1.50 \pm 0.07$, $P < 10^{-10}$; GPi, from $0.89 \pm 0.03$ to $0.99 \pm 0.03$, $P < 10^{-6}$). The sequential correlation (i.e., the correlation in spike count between two successive windows [window size, 20 ms]), which quantifies the temporal dependence of spikes[36], increased in the STN/GPe/GPi during the POST period (Fig. 4b; STN, from $0.061 \pm 0.037$ to $0.121 \pm 0.034$, $P < 10^{-4}$; GPe, from $0.299 \pm 0.020$ to $0.375 \pm 0.022$, $P < 10^{-9}$; GPi, from $0.172 \pm 0.016$ to $0.224 \pm 0.015$, $P < 10^{-6}$). The Fano factor, a measure of the variability in spike trains across trials, increased in the STN/GPe/GPi during the POST period (Fig. 4b; 100-ms bin size; STN, from $1.23 \pm 0.11$ to $1.57 \pm 0.14$, $P < 10^{-4}$; GPe, from $1.64 \pm 0.09$ to $2.23 \pm 0.13$, $P < 10^{-10}$; GPi, from $1.14 \pm 0.07$ to $1.31 \pm 0.07$, $P < 10^{-4}$). Time course of firing rate and firing pattern changes along with task performance were examined in monkeys K and U with DCZ administration

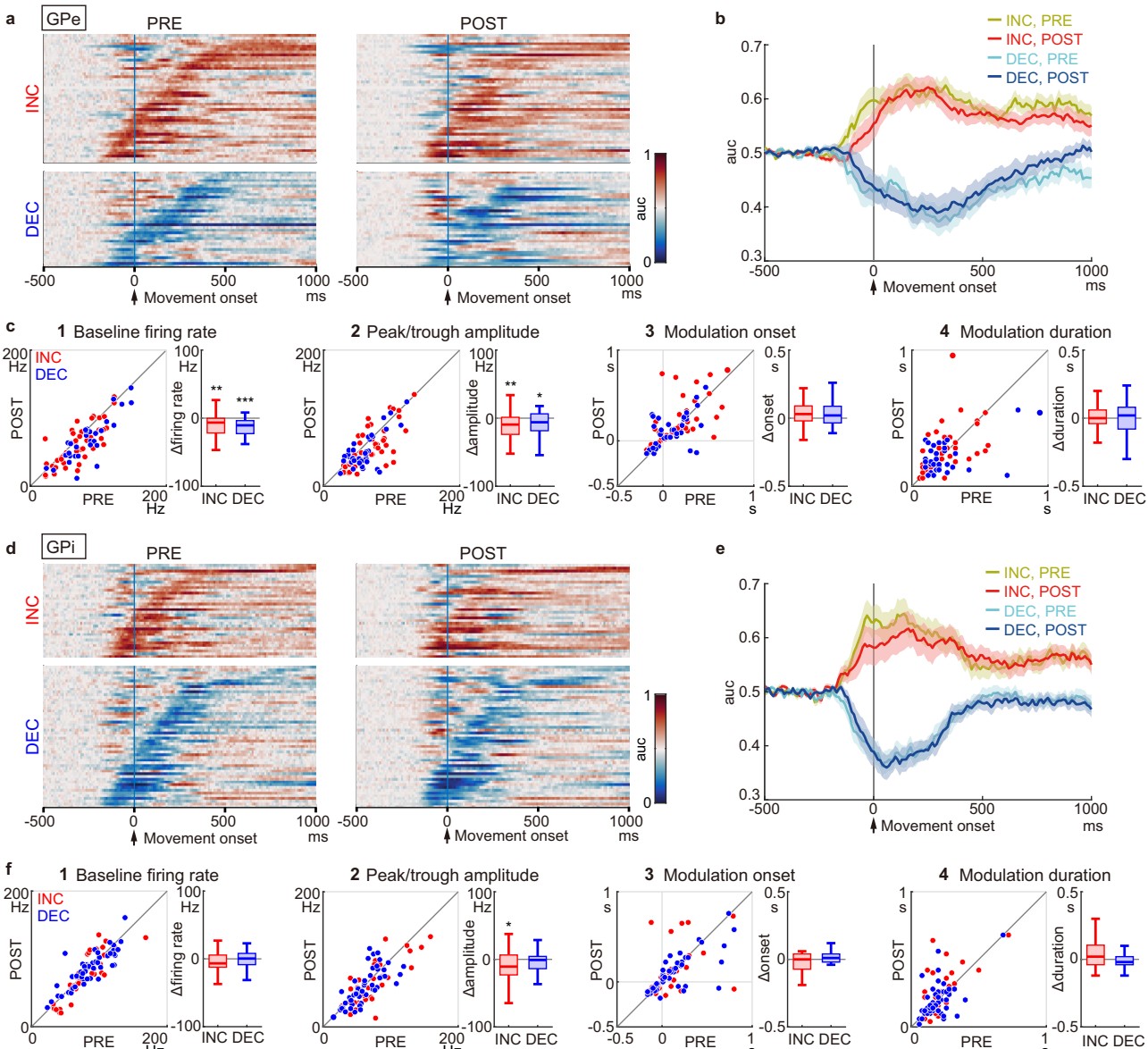

**Fig. 3 Changes in movement-related activity of GPe/GPi neurons after reduction of STN activity in ET trials. a** Heatmaps aligned with Movement onset for 78 GPe neurons classified as 43 INC- and 35 DEC-type neurons from monkeys K and U. Firing rates were converted to area under curve (auc) of ROC analysis. Neurons are sorted by the onset of movement-related activity in the PRE period (left). The activity of the same neuron in the POST period is shown on the same row (right). Bin width, 10 ms. **b** Population-averaged PETHs of INC- and DEC-type GPe neurons in the PRE and POST periods. **c** Scatter and box plots of change in PETHs between the PRE and POST periods of 43 INC- and 35 DEC-type neurons. **c1** Baseline firing rate during the 500 ms preceding Task cue decreased (INC, $P = 0.002$; DEC, $P = 1 \times 10^{-5}$; two-tailed Wilcoxon-signed rank test); **c2** Peak (INC) or trough (DEC) amplitude of the PETHs decreased (INC, $P = 0.004$; DEC, $P = 0.02$); **c3** Onset of movement-related modulations; **c4** Duration of movement-related modulations. *$P < 0.05$, **$P < 0.01$, ***$P < 0.001$. **d–f** Same as (**a–c**) but for 83 GPi neurons classified as 33 INC- and 50 DEC-type neurons from monkeys E, K, and U. **f2** Peak (INC) amplitude of the PETHs decreased ($P = 0.02$). Solid lines and shading indicate mean and SEM, respectively (**b**, **e**). In box plots, an inner horizontal line represents median; box, 25th and 75th percentiles; whiskers, maximum and minimum values within 1.5 times the interquartile range from the upper and lower quartiles (**c**, **f**).

(Supplementary Fig. 11). All firing pattern changes occurred within 1.0–4.5 min after DCZ administration and remained until the end of recordings (Supplementary Fig. 11a); no constant temporal order was found in changes between firing rate, pause probability, CV of ISIs, sequential correlation, and Fano factor in the STN/GPe/GPi. Behavioral changes tended to occur after neural activity changes (2.9–9.0 min after DCZ administration; Supplementary Fig. 11b), suggesting that the high spike train variability entrained a group of neurons, and then behaviors changed with time lag. We also examined oscillatory activity

changes by the power spectrum density function averaged across neurons after normalization with a local shuffling method (see Methods for details), and observed no oscillatory activity changes with reduction of STN activity (Fig. 4c).

To investigate the neural mechanism that contributed to the Fano factor increase, its correlations with the firing rate and sequential correlation were examined (Fig. 4d). The changes in the Fano factor were weakly correlated with changes in the firing rate in the STN/GPe/GPi (Fig. 4d *upper*; STN, $R^2 = 0.22$, $P < 0.01$; GPe, $R^2 = 0.15$, $P < 0.001$; GPi, $R^2 = 0.16$, $P < 0.001$). In contrast,

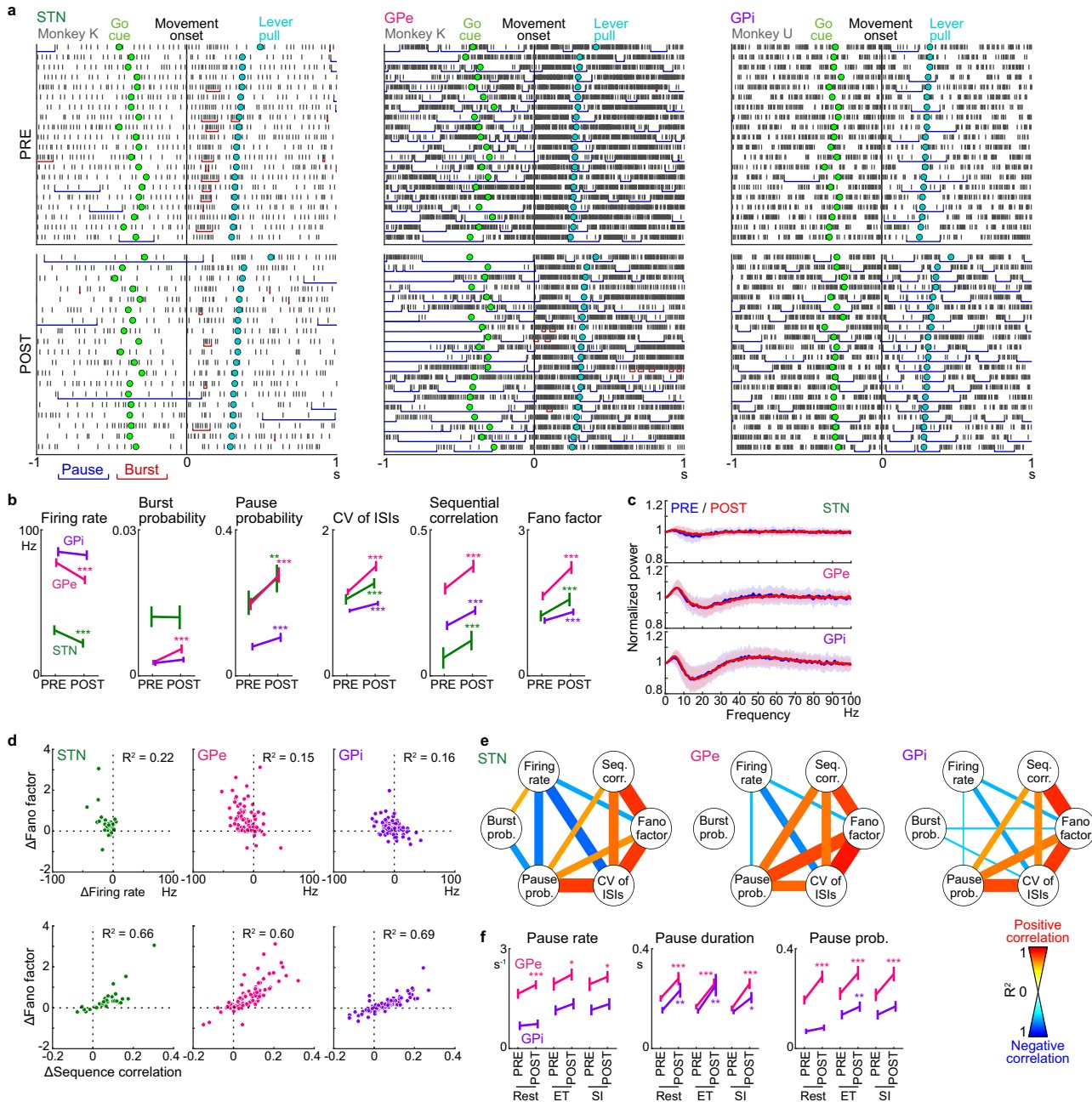

**Fig. 4 Firing pattern changes in the STN/GPe/GPi after reduction of STN activity. a** Examples of STN/GPe/GPi neuronal activity aligned with Movement onset and sorted by MTs in ET trials during the PRE and POST periods with bursts (red horizontal line) and pauses (blue). **b** Analyses of spike trains of 37 STN, 78 GPe, and 83 GPi neurons during −1.0 to 1.0 s relative to Movement onset in both ET and SI trials. Firing rate of STN and GPe neurons decreased ($P = 1 \times 10^{-6}$ and $2 \times 10^{-8}$, respectively; two-tailed Wilcoxon-signed rank test), while Burst probability of GPe neurons (0.0004), and Pause probability of STN/GPe/GPi neurons (0.004, $1 \times 10^{-8}$, 0.0001), CV of ISIs of STN/GPe/GPi neurons (0.0001, $1 \times 10^{-11}$, $6 \times 10^{-7}$), Sequential correlation of STN/GPe/GPi neurons ($7 \times 10^{-5}$, $3 \times 10^{-10}$, $4 \times 10^{-7}$), and Fano factor of STN/GPe/GPi neurons ($2 \times 10^{-5}$, $5 \times 10^{-11}$, $2 \times 10^{-5}$) increased. **c** Population-averaged power spectrum density (PSD) function. Solid lines and shading indicate means and SD. **d** Scatter plots representing changes of each neuron (POST−PRE) between Fano factor and Firing rate (above) and Sequential correlation (below). $R^2$, squared Pearson correlation. **e** Network representation for pairwise correlations between spike parameters in **b**. Width and color of links between nodes represent strength of correlations. **f** Analysis of pause in 78 GPe and 83 GPi neurons during the 0.5 s preceding Task cue (Rest) and from −0.2 to 0.3 s relative to Movement onset (ET and SI) in the PRE and POST periods. In the GPe, Pause rate during Rest, ET, and SI ($P = 5 \times 10^{-5}$, 0.02, 0.01; two-tailed Wilcoxon-signed rank test), Pause duration during Rest, ET, and SI ($1 \times 10^{-6}$, $1 \times 10^{-7}$, $5 \times 10^{-7}$), and Pause probability during Rest, ET, and SI ($2 \times 10^{-7}$, $2 \times 10^{-6}$, $4 \times 10^{-7}$) increased. In the GPi, Pause duration during Rest, ET, and SI (0.007, 0.008, 0.02) and Pause probability during ET (0.01) increased. Error bars indicate SEM. * $P < 0.05$, ** $P < 0.01$, *** $P < 0.001$.

the changes in the Fano factor were strongly and positively correlated with changes in the sequential correlation in all three nuclei (Fig. 4d *lower*; STN, $R^2 = 0.66$, $P < 10^{-9}$; GPe, $R^2 = 0.60$, $P < 10^{-16}$; GPi, $R^2 = 0.69$, $P < 10^{-22}$). The same analysis was applied to other statistical measures (Fig. 4e). In addition to the sequential correlation, changes in the CV of ISIs were positively correlated with Fano factor changes in all three nuclei. Interestingly, changes in the pause probability were positively correlated with Fano factor changes in the GPe/GPi but not the STN. The pauses in the GPe/GPi became more frequent and of longer duration both at rest and during movements in the POST period (Fig. 4f). These results suggest that reduction of STN activity induces sporadic pauses and interrupted spike trains in the GPe/GPi, resulting in highly irregular, unstable neural activity.

These changes observed in Figs. 3, 4 are consistent across animals either with CNO or DCZ administration. The PETHs and spike train variability were analyzed separately in each animal (Supplementary Fig. 12). GPe and STN neurons tended to decrease their baseline firing rates and peak/trough amplitude, while no changes were observed in modulation onsets and durations in the STN/GPe/GPi (Supplementary Fig. 12b, e, h). Pause probability, CV of ISIs, sequential correlation, and Fano factors in the STN/GPe/GPi were mostly increased in all three animals (Supplementary Fig. 12c, f, i). Furthermore, to examine possible off-target effects of CNO and DCZ, single-unit activity in the STN/GPe/GPi on the AAV non-injection side was recorded during the same task but using the hand contralateral to the recording side in monkey U, and no changes in baseline firing rates, PETHs, and firing pattern were observed with the administration of CNO or DCZ (Supplementary Fig. 13).

**Spike train variability and disturbed reaching movements**. Although reduction of STN activity increased spike train variability (Fig. 4) in the GPe/GPi, it is not clear how these changes disturbed reaching movements. Detailed observations of spike trains and MTs (Fig. 5a) suggested that the firing rate of STN neurons tended to be lower in trials with long MTs in both the PRE and POST periods and that the Fano factor of STN/GPe/GPi neurons tended to be higher in trials with long MTs. Thus, trials were grouped as short-MT trials (green, trials with MTs below the 25th percentile) or long-MT trials (magenta, trials with MTs above the 75th). Population-averaged PETHs revealed that the firing rate of the STN before Movement onset (from $-65$ ms) was significantly lower in the long-MT trials during the POST period, whereas no correlation between firing rate and MT was observed in the GPe/GPi (Fig. 5b, c). In contrast, population-averaged analysis revealed higher Fano factor in the long-MT trials during the POST period in the STN/GPe/GPi (Fig. 5d, e): from $-255$, $-165$, and $-185$ ms relative to Movement onset in the STN, GPe, and GPi, respectively. Interestingly, the Fano factor was higher in the long-MT trials in the STN/GPi during the PRE period as well, suggesting that reduction of STN activity exaggerates spike train variability observed in the normal state.

**Involuntary movements and phasic STN/GPe/GPi activity**. The relationship between phasic activity changes in STN/GPe/GPi neurons and involuntary movements was examined in the POST period (Fig. 6). Among neurons with a sufficient number of trials (>20) with involuntary movements, STN (17/37, 46%), GPe (18/78, 23%), and GPi (17/63, 27%) neurons exhibited significant firing rate modulations during the 200 ms preceding the onset of involuntary movements, as shown in Fig. 6a (data shuffling method with $\alpha = 0.05$, two-tailed; see Methods for details). In the GPe, DEC-type neurons during task performance tended to exhibit inhibition during

involuntary movements, and INC-type neurons tended to exhibit excitation (Fig. 6b; INC vs. DEC types, $\chi^2 = 7.7$, $P < 0.01$; chi-square test). A similar tendency was observed in the STN and GPi but not significant (STN, $\chi^2 = 1.3$, $P = 0.25$; GPi, $\chi^2 = 3.46$, $P = 0.06$). Both the excitation and inhibition preceded the onset of involuntary movements (Fig. 6c): excitation (STN, $-399 \pm 98$ ms; GPe, $-212 \pm 214$ ms; GPi, $-241 \pm 176$ ms; mean $\pm$ SD) and inhibition (STN, $-313 \pm 181$ ms; GPe, $-302 \pm 147$ ms; GPi, $-206 \pm 112$ ms).

**Discussion**

Reduction of STN activity increased spike train variability in STN/GPe/GPi neurons (Fig. 4). The mechanism of increased spike train variability is not clarified in the present study. GPe/GPi neurons lacked phasic firing rate increases at rest without phasic excitatory STN inputs[37], suggesting that GPe/GPi firing patterns are sensitive to STN inputs. Theoretical studies have shown that inputs to the GPe/GPi shift spike timings to desynchronize neural activity[38] and that the change of synaptic strength between the STN and GPe introduces noise through chaotic dynamics in the STN-GPe network[39]. Possibly, spike train variability is increased by higher sensitivity to external noise. Alternatively, the variable firing pattern in the STN may drive the increase of the spike train variability in the GPe/GPi, supported by the observation that the STN changed its firing pattern and rate in a similar time course (Supplementary Fig. 11a).

Exaggerated pauses in the GPe can also contribute to development of spike train variability (Fig. 4). The functional role of pauses in the GPe remains elusive, however. Many GPe neurons reportedly exhibit pauses in sub-human primates[34,40] and humans[41]. Although pauses were not correlated with any movements, relationships with alertness, task engagement, and motor learning have been reported[42–44]. Hence, the probability of pauses depends upon the animal's state, while the timing of individual pauses does not have any physiologic significance. Reduction in excitatory inputs from the STN to the GPe dramatically enhanced pauses (Fig. 4f), consistent with pharmacologic studies[8,35]. Assuming that the GPe affects the GPi either directly or indirectly, the random and irregular nature of pauses in the GPe is sufficient to impart variability to spike trains in the GPi.

The STN is believed to increase the baseline firing rate of the GPi, enhancing inhibition on the thalamus. The role of the STN in regulating movements, such as stopping or switching movements, was initially examined in DBS and imaging studies in humans[20–22], and later supported by electrophysiologic recordings in rodents[45]. Further studies showed that the STN neurons involved in movement stopping or switching are restricted to the ventromedial STN in sub-human primates[46,47] and humans[48], which is the target of the dorsolateral prefrontal cortex or pre-supplementary motor area[49,50]. The present study focused primarily on the dorsolateral STN (Fig. 1b and Supplementary Fig. 1), the target of the M1 and SMA[1,29,30,49,51]. Together with the motor-related STN activity observed during the reaching task (Supplementary Fig. 10), the dorsolateral STN plays a role more closely related to the movement itself rather than stopping or switching movements.

In the GPi, output nucleus of the BG, the baseline firing rate and movement-related activity were only weakly affected after reduction of STN activity (Fig. 3), while the spike train variability significantly increased (Fig. 4). Moreover, our across-trial analysis showed that long MT was associated with high spike train variability before Movement onset in STN/GPe/GPi neurons after reduction of STN activity, but no such trend was observed in terms of firing rate (Fig. 5). In long-MT trials, high spike train

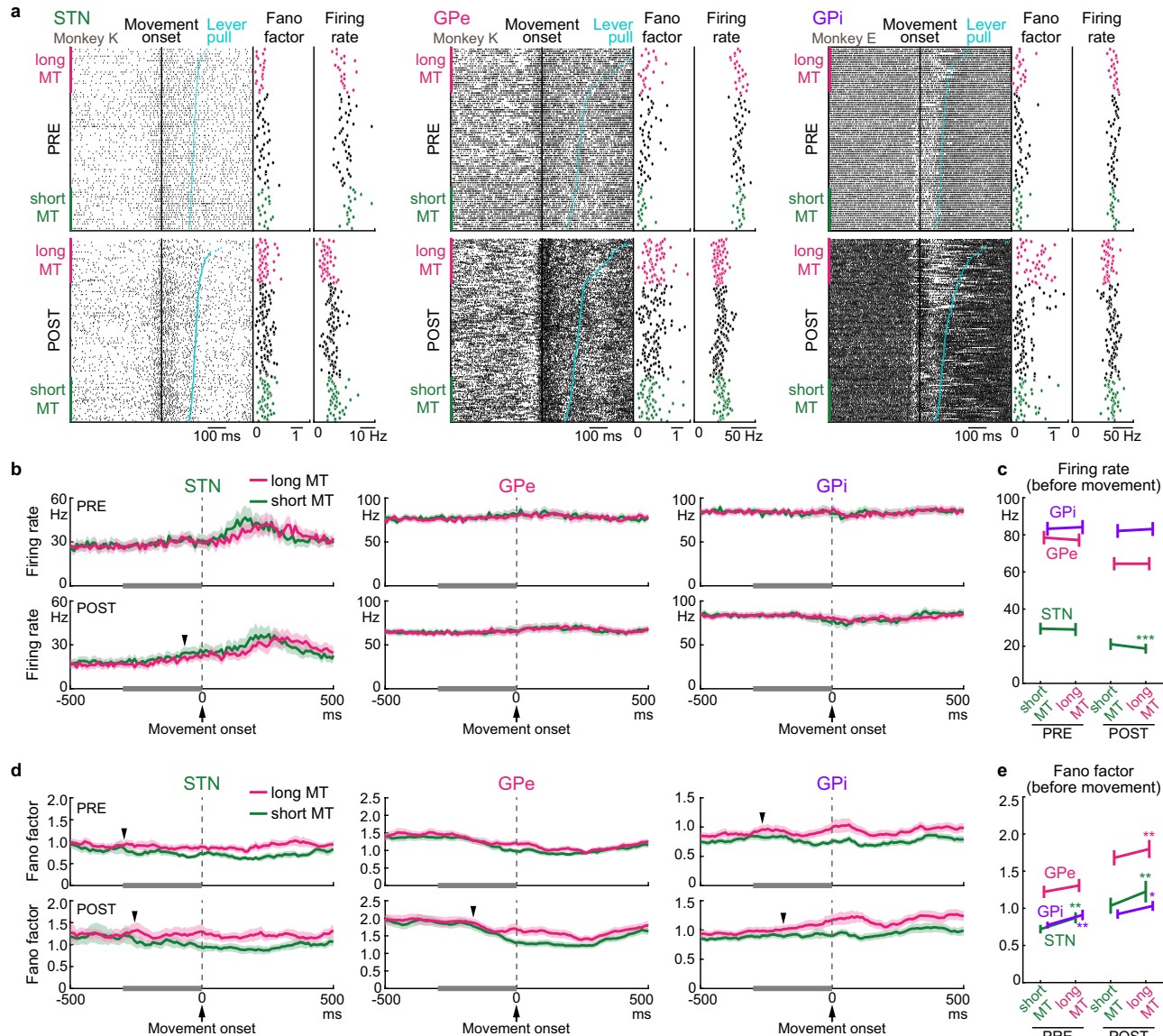

**Fig. 5 Neural activity correlated with disturbance in reaching movements after reduction of STN activity. a** Examples of STN/GPe/GPi neuronal activity with the Fano factor and Firing rate calculated in each trial. Spikes are aligned with Movement onset, and trials are sorted by MTs. In each neuron, the short- and long-MT trials were defined as trials with MTs below the 25th and above the 75th percentile, respectively. Both ET and SI trials were combined. **b** Population-averaged Firing rates of 35 STN, 78 GPe, and 80 GPi neurons (with ≥10 short- and long-MT trials) for the short- and long-MT trials in the PRE and POST periods. Solid lines and shading indicate mean and SEM, respectively. Arrowheads indicate onsets of significant changes between short- and long-MT trials. Bin width, 10 ms. **c** Firing rate difference of 35 STN, 78 GPe, and 80 GPi neurons between short- and long-MT trials during the 300 ms preceding Movement onset, indicated by gray horizontal lines in **b**. STN neurons showed low firing rate in long-MT trials of the POST period ($P = 0.0009$; two-tailed Wilcoxon-signed rank test). **d** Population-averaged Fano factor of 35 STN, 78 GPe, and 80 GPi neurons for the short- and long-MT trials in the PRE and POST periods. Solid lines and shading indicate mean and SEM, respectively. **e** Fano factor difference of 35 STN, 78 GPe, and 80 GPi neurons between short- and long-MT trials during the 300 ms preceding Movement onset. High Fano factor in long-MT trials in the STN and GPi ($P = 0.002$ and 0.009, respectively; two-tailed Wilcoxon-signed rank test) during the PRE, and in the STN, GPe, and GPi ($P = 0.002$, 0.007, and 0.01) during the POST. Error bars indicate SEM. *$P < 0.05$, **$P < 0.01$, ***$P < 0.001$.

variability in the STN preceded that in the GPe/GPi by 70-90 ms during the POST period (Fig. 5d). These observations suggest a causal relationship between reduction of STN activity, increased spike train variability in the GPe/GPi, and unstable movements. Based on these results, we would like to propose a function of the STN as follows (Supplementary Fig. 14). Excitatory STN inputs stabilize spike timing in GPe/GPi neurons, reduce trial-to-trial spike train variability during movements, and contribute to performance of rapid and stable movements. Reduction of such excitatory inputs increases spike train variability in GPe/GPi

neurons, and unstable GPe/GPi activity changes during movements disturb movements.

Compared with previous reports on interventions in the STN of normal sub-human primates (Supplementary Table 1), chemogenetic reduction of STN activity in the present study induced mild involuntary movements only in the forelimb and no firing rate changes in the GPi. This is presumably because reduction of STN activity was moderate (reduced neuronal activity by 25–35%) and limited only in its forelimb motor subregion. Abnormal movements were associated with firing rate changes in

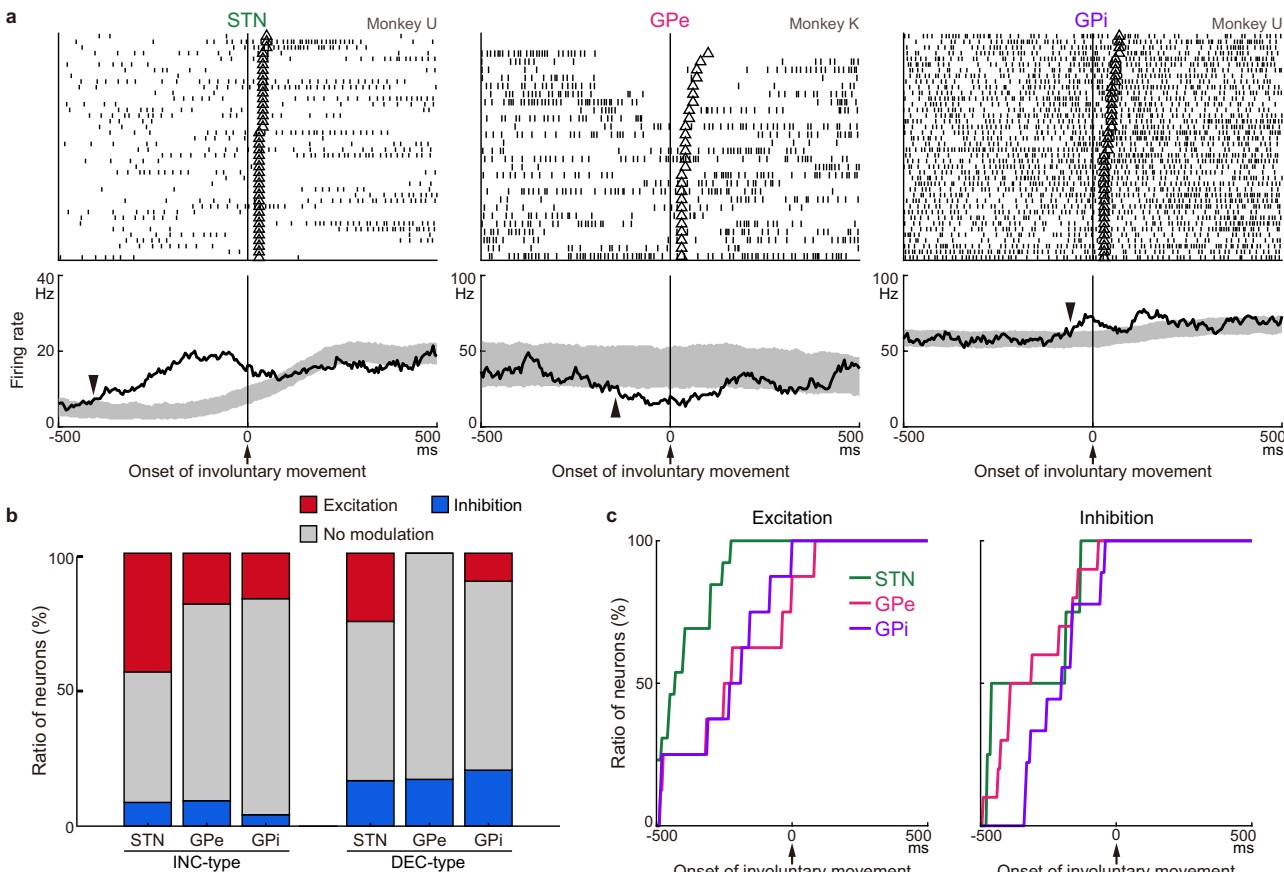

**Fig. 6 Neural activity in relation to involuntary movements after reduction of STN activity. a** Typical examples of STN/GPe/GPi neurons exhibiting excitation (STN and GPi) and inhibition (GPe) during involuntary movements. Spikes are aligned with the onset of involuntary movements, and trials are sorted by the duration of involuntary movements (The end of the involuntary movements was indicated by open triangles). In the PETHs (bin width, 5 ms; averaging window, 40 ms), solid lines and shading indicate averaged firing rates and 95% confidence interval estimated by a shuffling method, and the onset of activity modulation is indicated by arrowheads (−400, −135, and −50 ms for STN, GPe, and GPi neurons, respectively). **b** Ratios of STN/GPe/GPi neurons exhibiting significant excitation or inhibition before the occurrence of involuntary movements among 25 STN, 43 GPe, and 24 GPi INC-type neurons, and 12 STN, 35 GPe, and 39 GPi DEC-type neurons. **c** Cumulative histograms of the modulation onset timings for the STN/GPe/GPi neurons with excitation (left) or inhibition (right).

task-irrelevant timings in the GPi/GPe (Fig. 6). Neural activity in the STN/GPe/GPi exhibited similar modulation (i.e., increase or decrease) in involuntary movements and during the task, and such modulation well preceded the onset of involuntary movements. These results suggest that involuntary movements are induced via the following neural mechanism (Supplementary Fig. 14): (1) In the normal state, excitatory inputs from the STN stabilize GPe/GPi activity in the resting state and suppress involuntary movements; (2) Reduction of excitatory inputs from the STN increases spike train variability in the GPe/GPi; and (3) Coincident activity changes of GPe/GPi neurons, which are similar to those during voluntary movements, may occur and cause involuntary movements through the BG-thalamo-cortical pathway. However, the possible involvement of the projections from the BG, such as from the GPi and STN, to the brainstem motor centers could not be excluded. Further studies using pathway-specific manipulations of neural activity could test our hypothesis.

The concentration of DCZ in the cerebrospinal fluid is maximum at 90 min after administration[33], and clozapine, a presumed active ligand converted from CNO[52], would have higher concentration later (>90 min after administration)[53]. However, our electrophysiologic recordings could cover only 40–50 min after the ligand administration with good isolation and thus could

not observe their maximal effects. Consistent with the previous study[33], in our experiments, DCZ showed a faster time course than the CNO (Fig. 1d), but both CNO and DCZ exhibited similar behavioral and electrophysiologic effects after >20 min (Figs. 1, 2; Supplementary Fig. 12).

Loss of dopaminergic neurons in the substantia nigra pars compacta results in motor impairments in PD. Diminished transmission via the *direct* pathway and enhanced transmission via the *indirect* pathway lose movement-related inhibition in the GPi[5,54–56], resulting in akinesia; however, the baseline firing rate in the GPi does not necessarily increase[9,55–57]. The firing pattern also changes dramatically in PD; spike trains of many STN/GPe/GPi neurons exhibit correlated, oscillatory activity at the β frequency[54,58]. Both lesion/suppression and DBS of the STN exhibit therapeutic effects on PD symptoms;[15–17,55,59] these procedures may antagonize these GPe/GPi activity changes observed in PD. Chemogenetic manipulation in the present study moderately suppresses the STN activity and would also have beneficial effects on PD symptoms.

## Methods
**Experimental subjects.** Three Japanese monkeys (*Macaca fuscata*; E, male, 7.9 kg, 6 years old at the time of surgical operation; K, female, 6.7 kg, 7 years old; U, female, 5.1 kg, 4 years old) were used in this study. Each monkey was trained to sit

on the monkey chair. During behavioral experiments, access to drinking water was restricted and completely withheld for 24 h before experiments. We monitored their health conditions and maintained their body weight >90% of initial weight. The experimental protocols were approved by the Institutional Animal Care and Use Committee of the National Institutes of Natural Sciences. All experiments were conducted according to the guidelines of the National Institutes of Health *Guide for the Care and Use of Laboratory Animals*.

**Surgery**. Each monkey underwent surgical operation under aseptic conditions to fix its head painlessly in a stereotaxic frame, as previously described[8,60]. Under general anesthesia with ketamine hydrochloride (5−8 mg/kg body weight, i.m.), xylazine hydrochloride (0.5−1 mg/kg, i.m.), and propofol (5−7 μg/ml of target blood concentration, i.v.), the scalp was incised, the skull was widely exposed, and bolts made of polyether ether ketone (PEEK) or titanium were screwed into the skull as anchors. The skull was covered with bone adhesive resin (Super-Bond C&B, Sun Medical) followed by acrylic resin (Unifast II, GC Co). Two PEEK pipes were mounted in parallel over the frontal and occipital areas for head fixation. Antibiotics were injected (i.m.) after surgery.

After 1 week of recovery time, bipolar stimulating electrodes were chronically implanted to the motor cortices[1,8,29,60]. Under general anesthesia with ketamine hydrochloride (4–5 mg/kg, i.m.), the skull over the forelimb regions of the M1 and SMA was removed. To access the STN vertically or the GPe/GPi obliquely, the skull on the trajectories was also removed. The forelimb regions of the M1 and SMA were physiologically identified[8,60]. Three pairs of bipolar stimulating electrodes (200-μm stainless steel wires with Teflon coat; 2–2.5 mm inter-electrode distance) were then implanted in the distal and proximal forelimb regions of the M1 and the forelimb region of the SMA and fixed with acrylic resin. Rectangular plastic chambers were fixed to the skull with acrylic resin to access the STN and GPe/GPi. In monkey E, the stimulating electrode in the SMA became ineffective during experimental sessions, and only the stimulating electrodes in the M1 were used.

**Preparation of AAV**. The transfer plasmid, pAAV-CAG-hM4D-2a-GFP (Fig. 1a), was prepared from pAAV-CAG::FLEX-rev::hM4D-2a-GFP, a gift from Scott Sternson (Addgene plasmid #52536)[61]. The DNA fragment encoding hM4D-2a-GFP was separated at two *Eco*RI sites and inserted into the original plasmid in the inverted orientation, followed by the excision of the FLEX (loxP and lox2272) sequence at *Xba*I and *Spe*I sites.

The AAV vector was prepared as previously described[62]. Briefly, the plasmid vector was packaged with AAV-DJ capsid using the AAV Helper Free Expression System (Cell Biolabs); the packaging plasmids (pAAV-DJ and pHelper) as well as the transfer plasmid were transfected into HEK293T cells (American Type Culture Collection), which were harvested 72 h later and lysed by repeated freezing and thawing. The crude cell extract containing AAV particles was purified by ultracentrifugation with cesium chloride and concentrated by ultrafiltration using an Amicon 10 K MWCO filter (Merck Millipore). The copy number of the viral genome (vg) was $6.5–9.5 \times 10^{12}$ vg/ml, as determined using TaqMan Universal Master Mix II (Applied Biosystems).

**Mapping the STN**. Extracellular unit activity was recorded with glass-coated tungsten electrodes (1 MΩ, Alpha Omega) or homemade Elgiloy-alloy microelectrodes (0.5–1.5 MΩ at 1 kHz). A microelectrode was inserted vertically into the STN. Signals from the electrode were amplified, digitized at 44 kHz, digitally filtered between 0.5 and 9 kHz, and stored on a computer using a multi-channel recording system (AlphaLab SnR, Alpha Omega). A custom MATLAB script was used to manually isolate single-unit activity of the STN neurons. The motor subregion of the STN was identified based on (1) mid-frequency (30 Hz) firings, and (2) biphasic excitation to cortical stimulation (Fig. 1a; 0.3-ms duration; single pulse; intensity, 0.2 to 0.7 mA; inter-stimulus interval, 1.4 s) examined by constructing peri-stimulus time histograms (PSTHs) with 1-ms bins[1,8,29].

**Injection of AAV**. To precisely target the motor subregion of the STN, neural activity was recorded using a micropipette with a wire electrode when exploring the AAV injection sites. A glass micropipette was made from a borosilicate glass capillary (inner diameter, i.d., 1.8 mm; outer diameter, o.d., 3 mm.; G-3, Narishige) using a puller (PE-2, Narishige) and a beveler (EG-3, Narishige) and connected to a 25-μl Hamilton microsyringe (Hamilton Company) by a joint Teflon tube (JT-10, Eicom). A tungsten wire (30-μm core diameter with Teflon insulation; California Fine Wire Co.) was inserted into the micropipette to record neuronal activity. The glass micropipette, tubing, and Hamilton microsyringe were filled with mineral oil (MOLH-100, Kitazato Co.). The syringe was mechanically controlled by a syringe pump (IMS-20, Narishige). Viral vector solution was loaded from the micropipette. The glass micropipette was inserted vertically into the motor subregion of the STN through a small incision in the dura mater based on the STN mapping. After confirming the motor subregion of the STN by responses to cortical stimulation, 1 μl of the AAV solution was slowly infused at a rate of 0.05 μl/min. The micropipette was left in place for an additional 5 min and then slowly withdrawn. To cover the motor subregion of the STN, multiple injections were performed (1 μl per site, 1–2 sites per track, 1–2 tracks per day for 2–4 days); in total, 4, 15, and 21 μl were injected to the STN of monkeys E, K, and U, respectively.

**Reaching task**. As the BG may be differentially involved in the externally visual cued movements and the internally initiated movements without external cues[63,64], each monkey was trained to perform a custom reach-and-pull task, in which the monkey initiated the movements immediately after a "Go" cue presentation (externally triggered reaching trials, or ET trials) or without an apparent Go cue (self-initiated reaching trials, or SI trials; Fig. 2a). Task setup consisted of a home lever (2.5 cm in length, located 20 cm away and 25 cm below eye position), a full color LED (located at 25 cm away and 5 cm below), and a front lever (located at 22, 22, and 20 cm away for monkeys E, K, and U, respectively, and 7 cm below). First, the monkey sat on the monkey chair with its head fixed and pulled the home lever toward its body. After a random delay of 0.8–2.5 s, the LED turned on (Task cue) with a color instructing the trial type (blue, ET trial; red, SI trial). In ET trials, the LED color changed from blue to green (Go cue) in 1–2 s; the monkey was required to release its hand from the home lever (Movement onset) within 0.5 s and pull the front lever (Lever pull) within 3 s. In SI trials, the monkey was required to wait for 1.5 s but no more than 5 s, release its hand from the home lever (Movement onset), and pull the front lever (Lever pull) within 3 s. The trials were considered successful only if both the home lever release and the front lever pull were performed within the correct time windows. With a delay of 0.5 s after the front lever pull, the LED turned off; in a successful trial, 0.2 ml of juice was delivered as a reward. The two types of trials were randomly presented with an equal probability of 45%. The remaining 10% of trials were the same as SI trials except that the red LED turned green at 1.5 s (Instruction trials) to remind the monkey of the correct timing (i.e., 1.5 s after the red LED presentation) of the movement initiation for SI trials. The behavioral and electrophysiologic data obtained in Instruction trials were excluded from the analyses. The positions of the home and front levers were monitored using magnets attached to the levers and Hall-effect sensors fixed on the lever housings. Analog outputs from the sensors were recorded at 2750 Hz, down-sampled to 100 Hz, and converted to the lever positions.

RT was defined as the time from Go cue to Movement onset in ET trials and the time from Task cue to Movement onset in SI trials. MT was defined as the time from Movement onset to Lever pull. The behavioral task was controlled and logged using a custom script written in LabVIEW (LabVIEW 2013, National Instruments).

Stable performance was achieved in all monkeys after training for >3 months; the success rates were 95.1 ± 2.9, 88.7 ± 10.1, and 89.2 ± 10.5% in ET trials and 82.1 ± 10.1, 78.8 ± 7.4, and 91.6 ± 7.7% in SI trials for monkeys E, K, and U, respectively (mean ± SD; n = 21, 28, and 32 for monkeys E, K, and U, respectively). To avoid possible effects of DREADD ligands from the previous experiment, task sessions were performed every other day.

**Administration of CNO or DCZ**. CNO (HY-17366, MedChem Express) and DCZ (HY-42110, MedChem Express) were dissolved in dimethyl sulfoxide (DMSO) and then diluted with 0.9% saline to a final concentration of 1 mg/ml in 5% DMSO solution. DCZ is a newly developed ligand with high in vivo stability and high blood-brain-barrier permeability[33], without the potential off-target effects associated with CNO[52]. Aliquots were frozen at −20 °C for <2 weeks until used. The amount of DREADD ligand was determined in a pilot study to induce abnormal movements and used throughout the experiments: CNO, 1.0 mg/kg body weight (i.v.) and DCZ, 0.1 mg/kg body weight (i.m.). The PRE period was defined as from −15 to 0 min relative to administration of DREADD ligand. The POST period was defined as from 20 to 45 min for CNO administration and from 10 to 45 min for DCZ administration based on observations of STN activity (Fig. 1d), reflecting more rapid onset of DCZ than CNO. As a control, the same volume of VEH (5% DMSO in 0.9% saline) was administered (i.v. in monkey E; i.m. in monkeys K and U).

**Behavioral observation**. To reconstruct 3D trajectories for arm joints, RGB (x–y) and depth (z) images of the upper limb of the monkey during task performance were captured at 30 Hz using a depth camera (RealSense™ D435, Intel) and stored on a computer. To detect the positions of the arm joints, RGB (x–y) images were processed using DeepLabCut, as described previously[65,66], using a computer (Ubuntu 18.04, Intel Core i7-8750H and GeForce GTX 1050Ti). Training data were prepared by manually labeling the shoulder, elbow, wrist, and hand of 160 images from four task sessions, and the neural network was trained for 200,000 iterations. After determining the positions of the joints using a RGB (x–y) image, the distance from the camera to each joint was calculated as the average depth close to the corresponding position (≤10 pixels) in the depth image. Lastly, time series of 3D trajectories were constructed and digitally low-pass filtered (4th-order Butterworth, 7.5 Hz) before analysis. Trajectory deviation was defined as the total deviation from the mean trajectory; tortuosity was defined as the trajectory length divided by the distance between the start and end points. Speed of the joints was calculated from the displacement between two consecutive frames (1/30 s). Trajectory deviation and maximum speed were calculated during the Rest (from −1.0 to 0 s relative to Movement onset) and Movement (from 0 to 1.0 s) periods, and tortuosity during the period from −0.5 to 0.5 s.

**Recording of STN, GPe, and GPi activity**. The motor subregion in the GPe/GPi receiving input from the M1 and SMA was roughly mapped with extracellular unit recordings using the similar electrode for the STN mapping. The electrode was

obliquely (40° from vertical in the front plane) inserted, and spontaneous firing activity and cortically evoked responses were recorded. The GPe/GPi were identified based on (1) high-frequency (>60 Hz) firings, and (2) an excitation-inhibition-excitation triphasic response to cortical stimulation (the same parameters as those used for STN mapping)[8,29,35]. The GPe and GPi were distinguished by (1) the GPe-GPi boundary identified by silent zones corresponding to the medial medullary lamina; and (2) firing patterns: pauses observed in HFD-P GPe neurons but rarely seen in HFD GPi neurons[34,40].

Extracellular unit activity during task performance was recorded using 16-channel linear electrodes (0.8–2.0 MΩ at 1 kHz; inter-electrode distance, 100 or 150 µm; Plexon). The multi-channel electrode was inserted through a stainless steel guide tube (i.d., 0.45 mm; o.d., 0.57 mm) vertically into the STN or obliquely (40° from vertical) into the GPe/GPi. Signals from the electrodes were amplified, digitized at 44 kHz, digitally filtered between 0.5 and 9 kHz, and stored on a computer using a multi-channel recording system (AlphaLab SnR, Alpha Omega). In a total of 78 recording sessions, 1 to 5 well-isolated units (2.54 ± 1.11; mean ± SD) were simultaneously recorded. Cortically evoked responses (the same parameters as those used for STN/GPe/GPi mapping) of STN/GPe/GPi neurons were examined by constructing PSTHs with 1-ms bins, and neurons with cortical inputs were analyzed (Table 1). Cortical stimulation was performed every 5 or 10 min during the reaching task to examine the effect of the DREADD ligand (Supplementary Fig. 7). Electrophysiologic data obtained during cortical stimulation were excluded in analyses of baseline and movement-related activity.

**EMG recording.** EMGs of the biceps brachii and triceps brachii muscles were obtained using surface electrodes (NE-05, Nihon Kohden) for monkey K or chronically implanted stainless steel wire electrodes (7-stranded 25.4-µm diameter wire with Teflon coating; A-M Systems, Sequim, WA) for monkey U. Signals from the electrodes were amplified (5000×) and bandpass-filtered (150–3000 Hz) using an amplifier (MEG-5200, Nihon Kohden), digitized at 20 kHz (PCIe-6321, National Instruments), and stored on a computer. The root mean square (RMS) of each EMG was calculated with a 100-ms moving window. The RMS of the EMG was aligned with task events, and the mean and SEM were computed (Supplementary Fig. 5).

**Histology.** Monkeys E and K were sacrificed 58 and 89 weeks after AAV injection to examine viral transduction and confirm the sites of electrophysiologic recordings. Monkey U is still alive and used for further experiments. At the end of the experiments, electrolytic lesions were made with cathodal constant current (20 µA for 30 s) at the putative boundaries of the GPe/GPi. With an overdose of sodium thiopental (50 mg/kg, i.v.), the monkeys were perfused transcardially with 0.1 M phosphate buffer (PB) containing 10% formalin, followed by 10% sucrose in PB. The brains were removed, kept in 0.1 M PB containing 30% sucrose at 4 °C for cryoprotection, and serially cut with a freezing, sliding microtome (HM440E, Microm Co.) to obtain 50-µm-thick frontal brain sections.

For immunostaining, free-floating sections containing the STN/GPe/GPi were incubated with rabbit anti-GFP (1:1000; A11122, Invitrogen), mouse anti-NeuN (1:1000; MAB377, Millipore), and/or anti-glial fibrillary acidic protein (GFAP; 1:400; G3893, Sigma-Aldrich) primary antibodies overnight at 4 °C. The sections were then rinsed and incubated secondary antibodies for 1 h at room temperature: Alexa Fluor 488–conjugated goat anti-rabbit (1:500; A11043, Invitrogen), Alexa Fluor 594-conjugated goat anti-mouse (1:500; A11032, Invitrogen), and/or biotinylated donkey anti-rabbit (1:1000; 711-065-152, Jackson) antibodies. Biotin-labeled sections were further treated with avidin-biotin-peroxidase complex (PK-6100; Vector Laboratories) for 30 min, followed by 3, 3'-diaminobenzidine (DAB) with $H_2O_2$ for 5 min (Peroxidase Stain DAB Kit [Brown Stain]; Nacalai). The sections were mounted on a gelatin-coated slide glass with FluorSave reagent (Calbiochem) for fluorescent staining or Mount Quick (Daido Sangyo) for DAB staining. Sections were dehydrated with ethanol and xylene, with counterstaining each one of two adjacent sections with 1% Neutral Red (pH 5) for 5 min. Photographs were taken using an inverted microscope (BZ-X700, Keyence). Transduction efficacy in the STN was defined as the ratio of GFP-positive cells among NeuN-positive cells examined in every 6 sections containing the STN.

**Data analysis.** All data were analyzed using custom scripts written in MATLAB (MATLAB R2019b, MathWorks). Only successful trials were analyzed, except for the reaction and movement times shown in Fig. 2b and Supplementary Fig. 3a, b.

A custom MATLAB script was used to manually isolate single-unit activity of STN/GPe/GPi neurons. Quality of unit isolation was evaluated by calculating isolation score, the normalized distance between a spike cluster ($S_{cluster}$) and noise cluster[67]. Briefly, the Euclidean distance between two spikes ($X$, $Y \in S_{cluster}$) was normalized by distances to all other events (including both spikes and noise):

$$P_X(Y) \equiv \frac{\exp(-d(X, Y)(\lambda/d_0))}{\sum_{Z \neq X} \exp(-d(X, Z)(\lambda/d_0))} \qquad (1)$$

where $d(X, Y)$ is the Euclidean distance between events $X$ and $Y$, $\lambda$ is a constant, and $d_0$ is the average Euclidean distance in $S_{cluster}$. The sum of $P_X(Y)$ for all $Y$ in $S_{cluster}$, denoted as $P(X)$, would have a value ranging from 0 to 1 and indicate the

distance of $X$ to $S_{cluster}$ compared with noise:

$$P(X) \equiv \sum_{Y \in S_{cluster}, X \neq Y} P_X(Y) \qquad (2)$$

Then, isolation score was defined as the mean of $P(X)$ for all spikes in $S_{cluster}$:

$$\text{Isolation score} \equiv \frac{1}{N} \sum_{X \in S_{cluster}} P(X) \qquad (3)$$

where $N$ is the number of spikes in $S_{cluster}$. Isolation score ranges from 0 to 1, corresponding to poor and ideal isolation. In this study, $\lambda = 10$ was used. Isolation score was calculated every 30 s for all STN/GPe/GPi neurons, and only neurons with isolation score ≥0.6 during the whole recording period were used for analyses.

Behavioral effects of CNO or DCZ administration were similar across animals; in all three monkeys, involuntary movements at rest were observed in shoulder, elbow, wrist, and digits (Fig. 1e), and MT in ET and SI trials became longer (Fig. 2b). Thus, the datasets from all monkeys were combined for the electrophysiologic analyses in Figs. 3–6 and Supplementary Figs. 7, 8, and 10.

**Neuronal responses to cortical stimulation.** To examine the effect of reduction of STN activity, cortically evoked responses in STN/GPe/GPi neurons were analyzed (Supplementary Fig. 7). First, PSTHs with 1-ms bins were constructed for each neuron; the PSTHs were smoothed with a Gaussian distribution ($\sigma = 1.6$ ms) and transformed to z-scores with activity from the 100 ms preceding cortical stimulation. For GPe/GPi neurons, early excitation, inhibition, and late excitation were defined as from 3 to 200 ms after stimulation, whereas only early and late excitations were defined for STN neurons. Two consecutive bins with $z > 1.65$ or $z < -1.65$ were considered the onset of excitation or onset of inhibition, respectively. The latency of the response was defined as the time from stimulation to the onset, and the duration was defined as the period from the onset until the first bin of two consecutive bins within the threshold ($|z| \leq 1.65$). The amplitude was defined as the sum of the $|z|$ scores during each response. Early and late excitation was defined as excitation with latencies of <20 ms and ≥20 ms in the STN/GPe/GPi, respectively.

**Movement-related neuronal activity.** Raster plots of ET trials were constructed by aligning spikes at each task event, usually Movement onset (Supplementary Figs. 9 and 10a, b), and displayed chronologically before and after DREADD ligand administration. PETHs were constructed by averaging firing rate across all success trials (≥20 trials) in 10-ms bins in the PRE period (during the 15 min before DREADD ligand administration) and POST period (20–45 min after CNO administration or 10–45 min after DCZ administration). The spike activity during the 500 ms before the Task cue presentation was used to calculate the baseline firing rate. To quantify the movement-related modulation, a receiver operative characteristic (ROC) analysis was performed by using the pre-movement period (from −500 to −300 ms relative to Movement onset) as the baseline[68]. Briefly, the probability of having a number of spikes greater than $r$ was calculated in each 20-ms window, where $r$ varied from 0 to the maximum spike count of the neuron. Then, the ROC curve was plotted on an x–y plane as a curve parameterized by $r$: the probability during the baseline ($x$) versus that in each window ($y$). Area under curve (auc) below the ROC curve indicated an increase (auc > 0.5) or decrease (auc < 0.5) of firing rate, and the maximum and minimum of auc during the period from −300 to 1000 ms relative to Movement onset were used to classify neurons into the INC or DEC type. Modulation duration was defined as the duration that minimizes the P-value with the two-tailed Mann–Whitney U-test against the pre-movement period. The amplitudes of peaks and troughs were defined as the difference in peak and trough firing rates from the baseline firing rate, respectively. Heatmaps and population PETHs were constructed from auc values in the ROC analysis.

**Firing patterns.** Bursts and pauses in the ET and SI trials were detected using the Robust Gaussian Surprise method[69]. Briefly, 5-min spike trains were segmented from continuous recording including both ET and SI trials, but separately in the PRE and POST periods. The ISIs during the 5-min segment were log-transformed to give log(ISI)s. First, the central distribution of log(ISI) was calculated by excluding outliers, that is, bursts and pauses. The E-center was defined as the midpoint of the 5th and 95th percentiles of the log(ISI)s. The central set was defined as the log(ISI)s that fell within E-center ±1.64 × MAD, where MAD is the median absolute deviation of log(ISI)s. The Central Location 1 (C1-center) was defined as the median of the central set. The Central Location ($\mu$) was the median of the log(ISI)s that fell within C1-center ±1.64 × MAD. Then, normalized log(ISI)s (NLISIs) were defined as: $NLISI_i = \log(ISI_i) - \mu$. The distribution of NLISIs was assumed to be Gaussian, with $\sigma = 1.48 \times MAD$, and the P-value for each ISI was computed. ISIs below or above a statistical significance level ($P < 10^{-5}$) were defined as bursts and pauses, respectively. Bursts and pauses were extended to adjacent ISIs if their inclusion lowered the P-value.

To analyze the spike train variability in Fig. 4, the following parameters were calculated during the period from −1.0 to 1.0 s relative to Movement onset in both the ET and SI trials: firing rate, burst and pause probabilities (probabilities of a neuron being in bursts and pauses, respectively), pause rate (frequency of pause occurrence), pause duration (average durations of all pauses), CV of ISIs (standard deviation divided by the mean of the ISIs), sequential correlation (correlation of spike count between successive 20-ms windows in each trial), and the Fano factor

(variance divided by the mean number of spikes in a 100-ms window). Power spectrum density (PSD) of each neuron was normalized by a local shuffling method to compensate the effects of firing rates and refractoriness[14,70]. Briefly, PSDs were calculated from either the original digitized spike train (2-kHz sampling rate) or locally shuffled spike trains ($T = 50\text{–}70$ ms; $N = 50$) using Welch's method (non-overlapping Hann window with 4096 bins). Then, the original PSD was divided by the average of shuffled PSDs to obtain the normalized PSD.

Firing rates and Fano factors in the short- and long-MT trials were calculated separately between the PRE and POST periods in each neuron (Fig. 5). Then, population-averaged firing rates (Fig. 5b) and Fano factors (Fig. 5d) were calculated, and their differences between the long- and short-MT trials were compared using the two-tailed Wilcoxon-signed rank test (Fig. 5c, e). The onsets of significant changes were defined as the first consecutive 2 bins with $P < 0.05$ using the one-tailed Wilcoxon-signed rank test. Only neurons with enough short- and long-MT trials ($\geq 10$ trials each) were analyzed.

**Involuntary movements**. Spontaneous movements of each monkey were video-taped while a monkey was seated on a chair in a soundproof shielded room. Uncoordinated jerky movements observed on the contralateral shoulder, elbow, wrist, and digits were counted before (PRE) and 20–60 min after (POST) the DREADD ligand administration (Fig. 1e).

Involuntary movements of the upper limb were detected as the task-irrelevant movements of the home lever during the period from $-0.5$ to $1.0$ s relative to Task cue. First, home lever position during the period from $-0.5$ to $1.0$ s relative to Task cue in ET and SI trials was measured in each monkey during the POST period with VEH administration as the baseline, and the threshold was defined as the mean ± 3 SD. Then, in the POST period with DREADD ligand administration, the fluctuations of the home lever position exceeding the threshold were considered to be caused by involuntary movements. The onset and end of involuntary movements were defined as the first and last points exceeding the threshold, respectively. The home lever position during the period from $-0.5$ to $1.0$ s relative to Task cue was used to calculate the occurrence probability of involuntary movements (Fig. 2d, e and Supplementary Fig. 11b), and that during the period from $-1.5$ to $-0.1$ s relative to Movement onset was used to analyze neural activity related to involuntary movements (Fig. 6).

To analyze spike activity during the involuntary movements shown in Fig. 6, PETHs during the period of ±500 ms relative to the onset of involuntary movements were constructed with 5-ms bins and 40-ms averaging windows. Involuntary movements were observed at various task timings (Fig. 2d); the movement-related activity around Movement onset would affect the PETH. To exclude the effect of movement-related activity, shuffled PETHs were constructed from trials without any involuntary movements and compared with the original PETH. To construct a shuffled PETH, each trial in the original PETH was replaced with a 1-s segment of spike train without involuntary movements at the same timing relative to Movement onset. This process was repeated 1000 times to obtain 1000 shuffled PETHs. At each bin, the spike counts of the 1000 shuffled PETHs were sorted from the lowest to the highest. The confidence interval was defined as the spike count from the 25th to the 976th at each bin, corresponding to a significance level of 0.05. Neural activity was judged to be modulated if the total spike count during the 500 ms preceding the onset of involuntary movements was below the 25th (inhibition) or above the 976th (excitation) of the corresponding spike counts in the shuffled PETHs. Onset of activity modulation was defined as the beginning of $\geq 4$ bins ($\geq 20$ ms) exceeding the confidence interval during the period of ±500 ms relative to the onset of involuntary movements.

**Statistics**. The success rate, RT, and MT were calculated in each session, and the statistical significance was computed using the two-tailed Wilcoxon-signed rank test between the PRE and POST periods (Fig. 2b and Supplementary Fig. 3a, b). In the trajectory analysis, trials from two to three sessions were combined; trajectory deviation, tortuosity, and maximum speed were calculated in each trial and compared between the PRE and POST periods using the two-tailed Mann–Whitney U-test (Fig. 2c and Supplementary Fig. 3c). The occurrence probability of involuntary movements during the POST period was compared between sessions with DCZ administration and those with VEH using the two-tailed Mann–Whitney U-test (Fig. 2e).

The significance of decreases in STN activity was computed using the one-tailed Wilcoxon-signed rank test with Bonferroni correction (Fig. 1d). To determine the significance of changes in the PETHs (Fig. 3c, f, Supplementary Figs. 10e and 12b, e, h), firing patterns (Fig. 4b, f and Supplementary Fig. 12c, f, i), and neuronal responses to cortical stimulation (Supplementary Fig. 7c), the parameters of individual neurons were compared between the PRE and POST periods using the two-tailed Wilcoxon-signed rank test.

**Reporting summary**. Further information on research design is available in the Nature Research Reporting Summary linked to this article.

## Data availability
Data files for neural spike trains that support the findings of this study are available at Zenodo:[71] https://doi.org/10.5281/zenodo.6340403. Data files for behavioral tests and raw electrophysiologic signals are too large for an online repository, and they are available from the corresponding author upon request. The source data underlying Figs. 1d, e, 2b, c, e, 3c, f, 4b, d–f, 5c, e, 6b, c, Table 1 and Supplementary Figs. 1c, 2g, 3a–c, 7c, 10e, 11, 12b, c, e, f, h, i, 13b, c are provided as a Source Data file. Source data are provided with this paper.

## Code availability
Custom MATLAB scripts used to analyze the electrophysiologic signals and the 3D trajectory for arm joints are available at Zenodo:[72] https://doi.org/10.5281/zenodo.6346369.

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

## Acknowledgements

We thank S. Sato, H. Isogai, N. Suzuki, K. Awamura, K. Miyamoto, T. Sugiyama, and R. Kageyama for technical support; and Y. Yamagata for her critical reading of the manuscript. This work was supported by MEXT KAKENHI ("Non-linear Neurooscillology", 15H05873 to A.N.), JSPS KAKENHI (JP18K15340 to T.H., 19KK0193 to A.N., 16K07014 to S.C.), JST CREST (JPMJCR1853 to S.C.), AMED (JP18dm0307005 and JP21dm0207115 to A.N.), and Takeda Science Foundation (to T.H.) grants. Japanese monkeys used in the present study were obtained through the National Bio-Resource Project (NBRP) "Japanese Monkeys" of MEXT.

## Author contributions

T.H., S.C., and A.N. designed the study; K.K. generated the viral vector; T.H. and S.C. performed the experiments; T.H. analyzed the data; T.H., S.C., and A.N. wrote the manuscript.

## Competing interests

The authors declare no competing interests.
