## [Peer review file · Nature Communications]

REVIEWER COMMENTS

Reviewer #1 (Remarks to the Author):

Hasegawa et al. use chemogenetic manipulation of the macaque STN to investigate its effect on single unit activity in STN, GPe, and GPi as well as on movement parameters. They find the method to silence STN and to vary its firing pattern, leading to increase variation of discharge in GPe and GPi and prolonged movement time. The authors conclude that the STN stabilizes spiking in the basal ganglia and thereby movement.

The contribution of STN activity to basal ganglia dynamics and movement in primates is an important topic and of clinical relevance, e.g. for Parkinson's disease and deep brain stimulation. This form of STN silencing in macaques, in particular combined with recordings in GPe and GPi, is novel.

The presented data are of high quality and analyses are carefully explained. Nevertheless, I have several concerns affecting the validity of the drawn conclusions.

Major 1:

The authors describe rate as well as pattern changes in STN during the POST period, compared to the PRE period (Fig. 4b). Next to the firing rate decrease, also an increase in the CV of ISIs, in sequential correlations, and in the Fano factor in STN are reported. However, the authors interpret any changes in GPe and GPi as effects of STN *silencing*, i.e. rate decreases. A different hypothesis would be that pattern changes in STN drive the pattern changes in GPe and GPi. How do the authors disentangle these two hypotheses? Was there a fraction of simultaneous recordings in STN and GPe/GPi that would allow for analysis of paired STN-GPe/GPi recordings to approach this question?

Major 2:

Fig. 1b indicates EGFP, and therefore presence of the AAV vector, also outside of STN. Did the authors quantify the distribution of transducer neurons *outside* of STN in monkeys E and K? What is the confidence of specifically affecting STN independent of surrounding structures?

Major 3:

Movement related activity was analyzed based on baseline firing rates, with the onset of a modulation defined in terms of mean baseline firing rates and their standard deviation. As firing rates typically show a skewed distribution (firing rates below 0 Hz are not possible, but fairly high firing rates are), this

approach may lead to a bias, affecting both the fraction of increases/decreases as well as their respective timing.

Major 4:

The authors find drastic behavioral effects of chemogenetic intervention in STN, and rather slight changes in neural activity in GPi. Given that the GPi is described as the only output stage of the basal ganglia here (projecting to thalamus, as shown in Fig. 1), can the authors comment on/ discuss how the slight changes in GPi activity mechanistically may translate to thalamic/cortical activity and behavioral changes? Specifically, lines 310-312 state “Coincident pauses/spikes across neurons occur with an increased probability, leading to sporadic involuntary movements”. What is the authors’ hypothesis on how coincident pauses/spikes translate to involuntary movement?

Major 5:

For detection of bursts and pauses, were NLISIs computed separately for each condition (PRE/POST, REST/SI/ET), or for all data combined? Depending on this choice, a bias for the detection of bursts/pauses in some conditions may apply.

Minor 1:

I suggest that GPe should not be called “the relay [...] nucle[us] of the basal ganglia” (line 15), as it supposedly actively performs complex computations in the basal ganglia [e.g. Sadek et al., J Neurosci, 2007].

Minor 2:

In lines 314-316, the authors write: “These observations suggest that involuntary movements are conveyed through the same cortico-BG pathway that regulates normal voluntary movements.” As it is not known to what extent the measured activity actually conveys movement (involuntary or voluntary), I suggest to remove or reformulate this statement.

Reviewer #2 (Remarks to the Author):

Hasegawa et al describe studies in which a chemogenetic approach was used to reduce the activity of the subthalamic nucleus (STN) in macaques. Injections of viral vectors were used to express the inhibitory DREADD hM4Di in the STN unilaterally. The readouts of the DREADD activation (after systemic injections of either CNO or DCZ) included involuntary movements, alterations in the movement during

the execution of a reaching task, and recordings in the STN and in the external and internal globus pallidus (GPe and GPi respectively). The results show that the DCZ or CNO injections decreased the firing rate in the STN, induced involuntary movements in the body side opposite to the viral vector injection, increased the movement times in the reaching time, and increased the firing variability in GPe and GPi.

This is a novel study with very interesting and thoroughly analyzed results. Given the limited number of published papers using chemogenetics in non-human primates and the importance of extending the use of these techniques to primates, the field could greatly benefit from this report. To my knowledge, this is the first report of the use of chemogenetic tools to modulate the activity in the subthalamic nucleus in non-human primates and could provide a foundation for the use of chemogenetic methods to modulate the activity of basal ganglia circuits in normal and disease states in non-human primate models. The study is very comprehensive, including a detailed analysis of motor behavior, as well as an extensive analysis of firing rates and patterns of STN, GPe and GPi neurons in relation to the reaching task and in response to cortical stimulation. The enthusiasm for this manuscript is diminished due to some important considerations about the data presented. Specifically, the limited histology data provided, the lack of control experiments with CNO and DCZ and the lack of information about data originated from individual animals are concerning. These and other concerns are described in the following text.

Major comments:

1. Given that the efficiency of the DREADD approach depends, ultimately, on the expression of the receptors on the intended neurons, the histological analysis should be more complete. This analysis should include (for both monkeys):
 - a. Lower magnification images showing brain regions surrounding the STN to confirm that the GFP is only expressed in the STN and not in surrounding areas.
 - b. Higher magnification images in the STN, to illustrate the pattern of cell staining (i. e., is the expression found in cell bodies, dendrites, neuropil...?).
 - c. Additional experiments should be done to demonstrate that the GFP expression is limited to neurons (use markers of astrocytes and other glial cells to verify lack of GFP in non-neuronal cells). This is particularly important, given that DREADD expressed on astrocytes can modify neuronal activity. 1
 - d. Images should also be provided of areas that receive projections from the STN (GPe and GPi) to demonstrate lack of retrograde transport of the viral particles. It is very important to provide this information, given that GPe and GPi neurons showed changes in firing patterns after administration of DREADD actuators.
 - e. The drawings shown in supplementary figure 1 should be complemented with a quantification of proportion of neurons expressing GFP for both monkeys. The drawing should indicate the pattern of GFP-positive neurons found in each monkey.

2. Another concern about the histology is the fact that, so far, there is no histology data for monkey U. While it is understandable that the tissue for the third animal is not yet available, this brings some uncertainty to the data, particularly because it is unclear which data is coming from which animal (see below). A detailed description of the histology for animals E and K, demonstrating consistent expression in these 2 animals would reduce this concern.

3. The study does not include control experiments in which naïve animals (with no AAV injections) receive systemic injections of CNO or DCZ. These are indispensable controls to include in all DREADD studies. It has been reported that both CNO (which ultimately converts to clozapine to enter the CNS) and DCZ can have off-target effects on neuronal activities and behaviors in naïve animals (see for example ref. 2). To some extent this concern is reduced by the fact that after systemic injections of DCZ or CNO there were no effects on the success rate, reaction time and movement time in the ipsilateral side (supplementary fig. 2). However, similar control experiments are needed for the electrophysiological recordings.

4. Similarity of data across animals: The data from all 3 monkeys was pooled together based on the criteria that the single unit recordings were qualitatively similar in the pre-injection period for all animals (table 1). However, electrophysiology data for the STN and GPe was obtained only for two monkeys. A more appropriate criteria to pool them together would be that the behavioral responses to the systemic injections were similar among the 3 animals .

5. The individual data per animal is shown for the reduction in firing in STN (fig 1) and for the behavioral data in figure 2. The individual data for all electrophysiological recordings should be reported. This individual data should help answer the following questions:

a. Were the effects observed after CNO for monkey E similar as those observed after DCZ for monkey K and U?

b. For monkeys K and E (for which histology is available), is the expression of DREADDs correlated with the intensity of the responses?

6. In relation to point 5: figures where the responses of all neurons are shown (as in fig. 3 or supp. fig 6), there should be information for the reader to know which neurons are coming from which animal. Perhaps show supplementary figures breaking down the data by monkey).

7. Also related to point 5: In figures showing individual examples, indicate which monkey contributed the data (as in the examples shown in fig 4a and 5a).

8. Additional information is needed to verify stability of recording of STN neurons for 25 minutes (as shown in fig 1c). The following information should be included:

- a. At least for some example neurons, data should be provided showing original traces of the spike waveform before (-15 min) and after (25 min) of drug administration.
- b. Include an analysis to verify consistent cell isolation throughout the recording period (see, for example, the use of the isolation score calculated every 30 sec of recording in similar studies recently published, ref. 3)

9. The timing of the data collection after drug injections may have been suboptimal, according to the following rationale:

o In this study, behavioral and electrophysiological data were collected from 10 to 45 minutes after DCZ injections. Data by Nagai et al 2020 4 indicates that the maximal concentration of DCZ (after i.m. injections as used in this studies) in CSF is reached at 90 min post injections. The present study collected behavioral and electrophysiological data from 10 to 45 min (although it is noted that the involuntary movements lasted for >2 h)

o For CNO experiments, the data collection period was 20 to 45 min after injection. CNO reaches a maximal concentration in CSF at 80 min post injection⁵. As it is known, CNO converts to clozapine, which is the real actuator for DREADDs, and the levels of clozapine in CSF peak even at later time points⁶.

Thus, for both compounds used in this study it is possible that the data collection period ended too soon, and the maximal effects of the compounds were not observed. This should be acknowledged in the paper as a technical limitation, with suggestions on extended periods of observation for future DREADD-related experiments.

10. The description of involuntary movements outside of task is not clear. In lines 83-87 (and as shown in supplementary video 1), the involuntary movements are described to occur while the “monkeys sat quietly in a chair without performing any task”. This description remains anecdotal. Could these movements be quantified?

11. The definition of the involuntary movements during the reaching task is not fully clear either. This definition is based on the fluctuations of the home lever in the POST period after drug injections (lines 603-606: “In the POST period, the home lever position exceeding the mean \pm 3SD calculated from the same duration in the PRE period was considered to be caused by involuntary movements. The onset and end of involuntary movements were defined as the first and last points exceeding the mean \pm 3SD, respectively”). A possible confound is that in the POST period involuntary movements appear as a consequence of fatigue. The involuntary movements should probably be defined based on the “POST” trials of experiments in which a vehicle was administered.

Other comments:

1. Methods: infusion rate of 0.05nl/min seems exceedingly slow. If this is correct, 1 ul was infused in ~30 hours! Please check.

2. Are the differences shown in fig 5b and 5c significant?

3. Throughout the manuscript the term "STN suppression" should be toned down, since the activity in the STN was not really suppressed by the chemogenetic treatment. Consider "reduction of STN activity" (as used in the title) or something similar.

4. References needed after "...only a few applications in non-human primates are reported, and electrophysiologic evaluation at the single-neuron level is lacking" (lines 54-55)

5. Please describe in more detail why two types of SI conditions were used in the task. What is the purpose of the Si trials with instruction? Are these included in the stats for the SI trials?

6. In Table 1, clarify that the neurons reported in this table correspond to those that responded to M1 and/or SMA stimulation.

7. The Discussion includes a description about network transitions and their effects on spike variability. Can the authors cite reports of data analysis or computer simulations dealing specifically with the GPe-STN circuit. For example, the work of Charlie Wilson 7 or others may be relevant to this discussion.

8. "Loss of excitatory inputs form the STN" (line 297) seems an overstatement. Perhaps more appropriate to say reduction in excitatory inputs.

9. Lines 304-305: replace "65-75%" for "25-35%", to reflect the proportion of reduction observed.

10. The discussion around mechanisms of action of DBS (lines 343-346) is slightly out of context for this paper and, also, outdated. There is more recent evidence implicating many mechanisms that could account for the effects of STN-DBS. Consider updating (or perhaps removing) this part.

11. Legend to fig 6: “Ratios of STN/GPe/Gpi neurons exhibiting significant excitation or inhibition during involuntary movements...” Since the modulation occurs before the onset of the involuntary movement, perhaps may be better to say “Ratios of STN/GPe/Gpi neurons exhibiting significant excitation or inhibition before the occurrence of involuntary movements

Refs cited:

1 Yu, X., Nagai, J. & Khakh, B. S. Improved tools to study astrocytes. *Nature Reviews Neuroscience*, doi:10.1038/s41583-020-0264-8 (2020).

2 Upright, N. A. & Baxter, M. G. Effect of chemogenetic actuator drugs on prefrontal cortex-dependent working memory in nonhuman primates. *Neuropsychopharmacology*, doi:10.1038/s41386-020-0660-9 (2020).

3 Deffains, M. et al. In vivo electrophysiological validation of DREADD-based modulation of pallidal neurons in the non-human primate. *Eur J Neurosci*, doi:10.1111/ejn.14746 (2020).

4 Nagai, Y. et al. Deschloroclozapine, a potent and selective chemogenetic actuator enables rapid neuronal and behavioral modulations in mice and monkeys. *Nat Neurosci* 23, 1157-1167, doi:10.1038/s41593-020-0661-3 (2020).

5 Eldridge, M. A. et al. Chemogenetic disconnection of monkey orbitofrontal and rhinal cortex reversibly disrupts reward value. *Nat Neurosci* 19, 37-39, doi:10.1038/nn.4192 (2016).

6 Raper, J. et al. Metabolism and Distribution of Clozapine-N-oxide: Implications for Nonhuman Primate Chemogenetics. *ACS Chem Neurosci*, doi:10.1021/acschemneuro.7b00079 (2017).

7 Wilson, C. J. Active decorrelation in the basal ganglia. *Neuroscience* 250, 467-482, doi:10.1016/j.neuroscience.2013.07.032 (2013).

12.

Reviewer #3 (Remarks to the Author):

The manuscript “Subthalamic nucleus stabilizes movements by reducing neural spike variability in monkey basal ganglia: chemogenetic study” by Hasegawa and colleagues presents the behavioral and neurophysiological changes occurring following the injection of chemogenetic ligands to the motor part of the STN. This method was used for a partial suppression of STN activity and led to changes in neuronal firing and to abnormal movements which appear in-line with the previously shown hemiballism which resulted from complete suppression using methods such as lesions. The study is very

thorough and utilizes a method which is rarely implemented in monkeys demonstrating their great potential for exploring movement disorders.

Major comments:

1. The authors make a bold statement regarding the causal relation of the spike train variability and the movement stabilization. The data that they provide does not support this claim but rather demonstrates only a correlation between the two phenomena. It is unclear whether such a causal claim may even be potentially supported given the available tools and paradigms.
2. The monkeys display a very different motor state following the chemogenetic manipulation which manifests by different movements which are partially recorded. It is not clear that the variability the GPe/GPi activity is not an epiphenomenon of the changes in neuronal activity due to the movements themselves throughout the session.
3. There are multiple previous studies of the effect of lesions or chemical blocking of the STN and the formation of abnormal movements. It would be useful to explicitly compare (potentially also quantitatively) their behavioral and neurophysiological results in a table/graph format with the findings of the current study to emphasize the novel findings.
4. The assumption presented in the discussion that the variability is derived from network transitions is speculative as it does not seem to be supported by the results.

Minor comments:

1. Changes in the expression of oscillations, i.e. spectral analyses are not presented.
2. The relation of the presented results to isolated E-I networks is unclear as in the presented case the variability is highly affected by the "external noise" and not by the specific GPe-STN balance.
3. A few minor errors:

Page 6, Line 88: contralateral  ipsilateral

Page 15, line 273: marginally significant  not significant

Reviewer #1 (Remarks to the Author):

Hasegawa et al. use chemogenetic manipulation of the macaque STN to investigate its effect on single unit activity in STN, GPe, and GPi as well as on movement parameters. They find the method to silence STN and to vary its firing pattern, leading to increase variation of discharge in GPe and GPi and prolonged movement time. The authors conclude that the STN stabilizes spiking in the basal ganglia and thereby movement.

The contribution of STN activity to basal ganglia dynamics and movement in primates is an important topic and of clinical relevance, e.g. for Parkinson's disease and deep brain stimulation. This form of STN silencing in macaques, in particular combined with recordings in GPe and GPi, is novel.

The presented data are of high quality and analyses are carefully explained. Nevertheless, I have several concerns affecting the validity of the drawn conclusions.

Major 1:

The authors describe rate as well as pattern changes in STN during the POST period, compared to the PRE period (Fig. 4b). Next to the firing rate decrease, also an increase in the CV of ISIs, in sequential correlations, and in the Fano factor in STN are reported. However, the authors interpret any changes in GPe and GPi as effects of STN *silencing*, i.e. rate decreases. A different hypothesis would be that pattern changes in STN drive the pattern changes in GPe and GPi. How do the authors disentangle these two hypotheses? Was there a fraction of simultaneous recordings in STN and GPe/GPi that would allow for analysis of paired STN-GPe/GPi recordings to approach this question?

As the reviewer pointed out, high spike variability in the STN may have induced the firing pattern changes in the GPe/GPi. Unfortunately, the simultaneous recordings from STN-GPe/GPi neuronal pairs were not performed. Instead, the time course of firing rate and pattern changes with DCZ administration was examined (Supplementary Fig. 11). Since the firing rate and pattern of the STN changed in a similar time course (onset timings, 3-4.5 min after DCZ administration), we cannot exclude the possibility that the firing pattern change in the STN induced high spike variability in the GPe/GPi.

Hence, we added the following remarks in Discussion:

Reduction of STN activity increased spike variability in STN/GPe/GPi neurons (Fig. 4). The mechanism of increased spike train variability is not clarified in the

present study. (p17, lines 307-308)

Alternatively, the variable firing pattern in the STN may drive the increase of the spike train variability in the GPe/GPi, supported by the observation that the STN changed its firing pattern and rate in a similar time course (Supplementary Fig. 11). (p. 17, lines 312-314)

Major 2:

Fig. 1b indicates EGFP, and therefore presence of the AAV vector, also outside of STN. Did the authors quantify the distribution of transducer neurons *outside* of STN in monkeys E and K? What is the confidence of specifically affecting STN independent of surrounding structures?

Transduced cells outside of the STN were quantified for monkeys E and K (Supplementary Fig. 2). Also, the following statement was added to Results:

However, a small fraction of transduced cells were found dorsal to the STN such as the thalamus and zona incerta, presumably due to the leakage of AAV solution along the injection track (Supplementary Fig. 2a-c, g). (p. 5, lines 74-76)

Major 3:

Movement related activity was analyzed based on baseline firing rates, with the onset of a modulation defined in terms of mean baseline firing rates and their standard deviation. As firing rates typically show a skewed distribution (firing rates below 0 Hz are not possible, but fairly high firing rates are), this approach may lead to a bias, affecting both the fraction of increases/decreases as well as their respective timing.

As the reviewer pointed out, z-tests assume that samples are normally distributed and the population standard deviation is known. To exclude the possible bias, we applied receiver operative characteristic (ROC) analysis (Cohen et al., 2012), a nonparametric method, for movement-related activity of the STN/GPe/GPi and updated Fig. 3 and Supplementary Figs. 8 and 10. The details of the method are described in Method section (pp. 32-33, lines 622-631).

Major 4:

The authors find drastic behavioral effects of chemogenetic intervention in STN, and rather slight changes in neural activity in GPI. Given that the GPI is described as the only output stage of the basal ganglia here (projecting to thalamus, as shown in Fig. 1), can the authors comment on/ discuss how the slight changes in GPI activity

mechanistically may translate to thalamic/cortical activity and behavioral changes? Specifically, lines 310-312 state “Coincident pauses/spikes across neurons occur with an increased probability, leading to sporadic involuntary movements”. What is the authors’ hypothesis on how coincident pauses/spikes translate to involuntary movement?

To illustrate our hypothesis, we added Supplementary Figure 14. Also, we added the following description in Discussion and the figure legend of Figure 14:

Coincident pauses/spikes across GPi neurons could induce disinhibition in the thalamocortical activity and cause immature movements, resulting in involuntary movements (Supplementary Fig. 14). (p. 19, lines 359-361)

Spontaneous fluctuation of spike trains of GPe/GPi neurons in a resting state could induce the thalamocortical activity and cause immature movements, resulting in involuntary movements. (Figure legend of Supplementary Fig. 14)

Major 5:

For detection of bursts and pauses, were NLISIs computed separately for each condition (PRE/POST, REST/SI/ET), or for all data combined? Depending on this choice, a bias for the detection of bursts/pauses in some conditions may apply.

To detect bursts and pauses, NLISIs were calculated on every 5-min segment from continuous recording including REST/SI/ET conditions. Hence, PRE (-15 to 0 min) and POST (20 to 45 min, or 10 to 45 min) conditions were analyzed separately, but REST/SI/ET conditions were combined. To clarify this analytical step, we added the following statement in Methods:

Bursts and pauses in the ET and SI trials were detected using the Robust Gaussian Surprise method⁶⁷. Briefly, 5-min spike trains were segmented from continuous recording including both ET and SI trials, but separately in the PRE and POST periods. (p. 33, lines 637-639)

Minor 1:

I suggest that GPe should not be called “the relay [...] nucle[us] of the basal ganglia” (line 15), as it supposedly actively performs complex computations in the basal ganglia [e.g. Sadek et al., J Neurosci, 2007].

We modified the description in Abstract:

The subthalamic nucleus (STN) projects to the external (GPe) and internal (GPi) pallidum, the modulatory and output nuclei of the basal ganglia (BG), respectively, and ... (p. 5, lines 14-15)

Minor 2:

In lines 314-316, the authors write: “These observations suggest that involuntary movements are conveyed through the same cortico-BG pathway that regulates normal voluntary movements.” As it is not known to what extent the measured activity actually conveys movement (involuntary or voluntary), I suggest to remove or reformulate this statement.

According to the reviewer’s recommendation, we removed the statement regarding to the neural pathway for voluntary and involuntary motor control.

Reviewer #2 (Remarks to the Author):

Hasegawa et al describe studies in which a chemogenetic approach was used to reduce the activity of the subthalamic nucleus (STN) in macaques. Injections of viral vectors were used to express the inhibitory DREADD hM4Di in the STN unilaterally. The readouts of the DREADD activation (after systemic injections of either CNO or DCZ) included involuntary movements, alterations in the movement during the execution of a reaching task, and recordings in the STN and in the external and internal globus pallidus (GPe and GPi respectively). The results show that the DCZ or CNO injections decreased the firing rate in the STN, induced involuntary movements in the body side opposite to the viral vector injection, increased the movement times in the reaching time, and increased the firing variability in GPe and GPi.

This is a novel study with very interesting and thoroughly analyzed results. Given the limited number of published papers using chemogenetics in non-human primates and the importance of extending the use of these techniques to primates, the field could greatly benefit from this report. To my knowledge, this is the first report of the use of chemogenetic tools to modulate the activity in the subthalamic nucleus in non-human primates and could provide a foundation for the use of chemogenetic methods to modulate the activity of basal ganglia circuits in normal and disease states in non-human primate models. The study is very comprehensive, including a detailed analysis of motor behavior, as well as an extensive analysis of firing rates and patterns of STN, GPe and GPi neurons in relation to the reaching task and in response to cortical stimulation. The enthusiasm for this manuscript is diminished due to some important considerations about the data presented. Specifically, the limited histology data provided, the lack of control experiments with CNO and DCZ

and the lack of information about data originated from individual animals are concerning. These and other concerns are described in the following text.

Major comments:

1. Given that the efficiency of the DREADD approach depends, ultimately, on the expression of the receptors on the intended neurons, the histological analysis should be more complete. This analysis should include (for both monkeys):

a. Lower magnification images showing brain regions surrounding the STN to confirm that the GFP is only expressed in the STN and not in surrounding areas.

We added Supplementary Figure 2; transduced cells in and around the STN were quantified for monkeys E and K. We found GFP-expressing cells outside of the STN, so the following description was added to Results:

However, a small fraction of transduced cells were found dorsal to the STN such as the thalamus and zona incerta, presumably due to the leakage of AAV solution along the injection track (Supplementary Fig. 2a-c, g). (p. 5, lines 74-76)

b. Higher magnification images in the STN, to illustrate the pattern of cell staining (i. e., is the expression found in cell bodies, dendrites, neuropil...?).

We added Supplementary Figure 2c, f showing magnified images of GFP expression in cell bodies and axonal terminals. We added the following description in the figure legend of Supplementary Figure 2.

EGFP expression was found in the cell bodies of the STN (c) and axonal terminals in the GPe/GPi (f, arrows). (Figure legend of Supplementary Fig. 2)

However, since the inhibitory DREADD receptor, hM4Di, was presumably cleaved from GFP by 2A self-cleaving peptide, hM4Di could be differently localized within a cell. No antibody specific to hM4Di is available and hM4Di was not designed to have an epitope tag (such as a hemagglutinin tag); hence, intracellular localization of hM4Di could not be identified in our design.

c. Additional experiments should be done to demonstrate that the GFP expression is limited to neurons (use markers of astrocytes and other glial cells to verify lack of GFP in non-neuronal cells). This is particularly important, given that DREADD expressed on astrocytes can modify neuronal activity. 1

We added Supplementary Figure 1a, b, where the selectivity of GFP expression was examined using double immunostaining with GFP and NeuN, or GFP and glial fibrillary

acidic protein (GFAP).

We described the results as follows:

Their colocalization, defined as the ratio of NeuN-positive cells among all GFP-positive cells, was 90%. **b**, Same as **(a)** but for astrocytes using anti-GFP (green) and anti-GFAP (magenta) antibodies. Colocalization of GFP and GFAP was 7%. (Figure legend of Supplementary Fig. 1).

Histologic examination of monkeys E and K revealed that EGFP expression was found in the dorsolateral part of the posterior STN (Fig. 1b), corresponding to the motor subregion^{24,30-32}, with preference to neurons (Supplementary Fig. 1a, b) ... (p. 5, lines 70-72)

d. Images should also be provided of areas that receive projections from the STN (GPe and GPi) to demonstrate lack of retrograde transport of the viral particles. It is very important to provide this information, given that GPe and GPi neurons showed changes in firing patterns after administration of DREADD actuators.

We added Supplementary Figure 2e, f showing images of the GPe/GPi. We added following description in Results:

No sign of retrograde transduction was found in the GPe/GPi (Supplementary Fig. 2d-f, g). (p. 5, lines 76-77)

e. The drawings shown in supplementary figure 1 should be complemented with a quantification of proportion of neurons expressing GFP for both monkeys. The drawing should indicate the pattern of GFP-positive neurons found in each monkey.

We added Supplementary Figure 1c *right*. The ratio of GFP-positive neurons was quantified in monkeys E and K

2. Another concern about the histology is the fact that, so far, there is no histology data for monkey U. While it is understandable that the tissue for the third animal is not yet available, this brings some uncertainty to the data, particularly because it is unclear which data is coming from which animal (see below). A detailed description of the histology for animals E and K, demonstrating consistent expression in these 2 animals would reduce this concern.

We added Supplementary Figures 1c and 2g for histologic data, showing that the transduction efficiency and distribution of transduced cells were similar between monkeys E and K. Also, the following statement was added in Results:

Histologic examination of monkeys E and K revealed that EGFP expression was found in the dorsolateral part of the posterior STN (Fig. 1b), corresponding to the motor subregion^{24,30-32}, with preference to neurons (Supplementary Fig. 1a, b) and similar transduction efficiencies in both monkeys (60.2% and 68.3% of neurons in monkeys E and K, respectively; Supplementary Fig. 1c). (p. 5, lines 70-74)

3. The study does not include control experiments in which naïve animals (with no AAV injections) receive systemic injections of CNO or DCZ. These are indispensable controls to include in all DREADD studies. It has been reported that both CNO (which ultimately converts to clozapine to enter the CNS) and DCZ can have off-target effects on neuronal activities and behaviors in naïve animals (see for example ref. 2). To some extent this concern is reduced by the fact that after systemic injections of DCZ or CNO there were no effects on the success rate, reaction time and movement time in the ipsilateral side (supplementary fig. 2). However, similar control experiments are needed for the electrophysiological recordings.

We performed an additional experiment in monkey U to examine off-target effects of CNO and DCZ on neural activity in the STN/GPe/GPi (Supplementary Fig. 13). No changes were observed as described in Results:

Furthermore, to examine possible off-target effects of CNO and DCZ, single unit activity in the STN/GPe/GPi on the AAV non-injection side was recorded during the same task but using the hand contralateral to the recording side in monkey U, and no changes in baseline firing rates, PETHs, and firing pattern were observed with the administration of CNO or DCZ (Supplementary Fig. 13). (p. 15, lines 272-275)

4. Similarity of data across animals: The data from all 3 monkeys was pooled together based on the criteria that the single unit recordings were qualitatively similar in the pre-injection period for all animals (table 1). However, electrophysiology data for the STN and GPe was obtained only for two monkeys. A more appropriate criteria to pool them together would be that the behavioral responses to the systemic injections were similar among the 3 animals.

Since behavioral changes were qualitatively similar across animals, we reasoned the electrophysiological data can be combined for analyses. The following description was added in Methods:

Behavioral effects of CNO or DCZ administration were similar across animals; in

all three monkeys, involuntary movements at rest were observed in shoulder, elbow, wrist, and digits (Fig. 1e), and MT in ET and SI trials became longer (Fig. 2b). Thus, the datasets from all monkeys were combined for the electrophysiologic analyses in Figures 3-6 and Supplementary Figures 7, 8, 10, and 11. (p. 31, lines 598-602)

5. The individual data per animal is shown for the reduction in firing in STN (fig 1) and for the behavioral data in figure 2. The individual data for all electrophysiological recordings should be reported. This individual data should help answer the following questions:

- a. Were the effects observed after CNO for monkey E similar as those observed after DCZ for monkey K and U?
- b. For monkeys K and E (for which histology is available), is the expression of DREADDs correlated with the intensity of the responses?

We analyzed neural activity separately in each animal (Supplementary Fig. 12). The spike train variability (i.e., CV of ISIs, sequential correlation, the Fano factor) increased in all three animals for the STN/GPe/GPi with both CNO and DCZ administration. The following description was added in Results:

These changes observed in Figures 3 and 4 are consistent across animals either with CNO or DCZ administration. The PETHs and spike train variability were analyzed separately in each animal (Supplementary Fig. 12). GPe and STN neurons tended to decrease their baseline firing rates and peak/trough amplitude, while no changes were observed in modulation onsets and durations in the STN/GPe/GPi (Supplementary Fig. 12b, e, h). Pause probability, CV of ISIs, sequential correlation, and Fano factors in the GPe/GPi were increased in all three animals (Supplementary Fig. 12c, f, i). (p. 14, lines 266-271)

Since the efficiencies and areas of transduction in the STN were similar between monkeys E and K (Supplementary Fig. 1c *right*), the relationship between electrophysiologic effects and transduction efficiency cannot be inferred from the two monkeys.

6. In relation to point 5: figures where the responses of all neurons are shown (as in fig. 3 or supp. fig 6), there should be information for the reader to know which neurons are coming from which animal. Perhaps show supplementary figures breaking down the data by monkey).

Supplementary Figure 12 was added to show neural activity change separately for each

animal.

7. Also related to point 5: In figures showing individual examples, indicate which monkey contributed the data (as in the examples shown in fig 4a and 5a).

All examples of neural activity were denoted with the identity of monkey from which neural activity was recorded.

8. Additional information is needed to verify stability of recording of STN neurons for 25 minutes (as shown in fig 1c). The following information should be included:

a. At least for some example neurons, data should be provided showing original traces of the spike waveform before (-15 min) and after (25 min) of drug administration.

b. Include an analysis to verify consistent cell isolation throughout the recording period (see, for example, the use of the isolation score calculated every 30 sec of recording in similar studies recently published, ref. 3)

We added Fig. 1c as an example STN neuron with raw extracellular signals, firing rates, and isolation score, which quantifies the quality of unit isolation. We re-analyzed all the neural activity and used only well-isolated units as described in Methods:

Isolation score was calculated every 30 s for all STN/GPe/GPi neurons, and only neurons with isolation score ≥ 0.6 during the whole recording period were used for analyses. (p. 31, lines 592-594)

9. The timing of the data collection after drug injections may have been suboptimal, according to the following rationale:

o In this study, behavioral and electrophysiological data were collected from 10 to 45 minutes after DCZ injections. Data by Nagai et al 2020 4 indicates that the maximal concentration of DCZ (after i.m. injections as used in this studies) in CSF is reached at 90 min post injections. The present study collected behavioral and electrophysiological data from 10 to 45 min (although it is noted that the involuntary movements lasted for >2 h)

o For CNO experiments, the data collection period was 20 to 45 min after injection. CNO reaches a maximal concentration in CSF at 80 min post injection⁵. As it is known, CNO converts to clozapine, which is the real actuator for DREADDs, and the levels of clozapine in CSF peak even at later time points 6.

Thus, for both compounds used in this study it is possible that the data collection period ended too soon, and the maximal effects of the compounds were not observed. This

should be acknowledged in the paper as a technical limitation, with suggestions on extended periods of observation for future DREADD-related experiments.

Based on the reviewer's suggestion, we added the following remark in Discussion:

The concentration of DCZ in the cerebrospinal fluid is maximum at 90 min after administration³³, and a clozapine, a presumed active ligand converted from CNO⁵¹, would have the highest concentration later. However, our electrophysiologic recordings could cover only 40-50 min after the ligand administration with good isolation and thus could not observe their maximal effects. (p. 19, lines 362-365)

10. The description of involuntary movements outside of task is not clear. In lines 83-87 (and as shown in supplementary video 1), the involuntary movements are described to occur while the “monkeys sat quietly in a chair without performing any task”. This description remains anecdotal. Could these movements be quantified?

We quantified involuntary movements in Figure 1e. Also, the following description was added to Results:

In the present study, administration of DREADD ligands often induced involuntary movements in all three monkeys, i.e., choreic/ballistic movements of the shoulder, elbow, wrist, and fingers contralateral to the AAV injection side while the monkeys sat quietly in a chair without performing any task (Supplementary Video 1; Fig. 1e). (p. 6, lines 89-92)

11. The definition of the involuntary movements during the reaching task is not fully clear either. This definition is based on the fluctuations of the home lever in the POST period after drug injections (lines 603-606: “In the POST period, the home lever position exceeding the mean \pm 3SD calculated from the same duration in the PRE period was considered to be caused by involuntary movements. The onset and end of involuntary movements were defined as the first and last points exceeding the mean \pm 3SD, respectively”). A possible confound is that in the POST period involuntary movements appear as a consequence of fatigue. The involuntary movements should probably be defined based on the “POST” trials of experiments in which a vehicle was administered.

We changed the method to detect involuntary movements such that the fluctuations of the home lever position in the POST period of control sessions was used as the baseline (Fig. 1d and Fig. 6). The following description was added to Methods:

First, home lever position during the period from -0.5 to 1.0 s relative to Task

cue in ET and SI trials was measured in each monkey during the POST period with vehicle administration as the baseline, and the threshold was defined as the mean \pm 3SD. Then, in the POST period with DREADD ligand administration, the fluctuations of the home lever position exceeding the threshold were considered to be caused by involuntary movements. (p. 35, lines 676-681)

Involuntary movements with DCZ administration were significantly higher than those with vehicle administration (Fig. 2e), excluding the possibility that abnormal movements were caused by muscle fatigue or motivational change.

Other comments:

1. Methods: infusion rate of 0.05nl/min seems exceedingly slow. If this is correct, 1 ul was infused in ~30 hours! Please check.

We fixed the infusion rate of AAV to be 0.05 μ l/min.

2. Are the differences shown in fig 5b and 5c significant?

We added statistical analyses between short- and long-MT trials during the 300 ms before Movement onset (Fig. 5b-e).

3. Throughout the manuscript the term “STN suppression” should be toned down, since the activity in the STN was not really suppressed by the chemogenetic treatment. Consider “reduction of STN activity” (as used in the title) or something similar.

Following the reviewer’s recommendation, “STN suppression” was replaced with “reduction of STN activity” throughout the manuscript.

4. References needed after “...only a few applications in non-human primates are reported, and electrophysiologic evaluation at the single-neuron level is lacking” (lines 54-55)

We modified the statement and referred a paper by Deffains, M. *et al.* (2020):

Although DREADDs have been utilized widely in rodents, application in sub-human primates is rather limited and only one electrophysiologic evaluation at the single-neuron level²⁸ is reported. (p. 4, lines 54-56)

5. Please describe in more detail why two types of SI conditions were used in the task. What is the purpose of the Si trials with instruction? Are these included in the stats for the SI trials?

To clarify the purpose of Instruction trials and analytical step, the following description was added in Methods:

The remaining 10% of trials were the same as SI trials except that the red LED turned green at 1.5 s (Instruction trials) to remind the monkey of the correct timing (i.e., 1.5 s after the red LED presentation) of the movement initiation for SI trials. The behavioral and electrophysiological data obtained in Instruction trials were excluded from the analyses. (p. 25, lines 472-476)

6. In Table 1, clarify that the neurons reported in this table correspond to those that responded to M1 and/or SMA stimulation.

We added numbers of neurons responded to only M1, both M1 and SMA, and only SMA stimulation in Table 1.

7. The Discussion includes a description about network transitions and their effects on spike variability. Can the authors cite reports of data analysis or computer simulations dealing specifically with the GPe-STN circuit. For example, the work of Charlie Wilson 7 or others may be relevant to this discussion.

We cited theoretical studies (Wilson, 2013; Kalva et al., 2012) that propose possible neural mechanisms to induce spike train variability in the GPe and/or GPi in Discussion:

The mechanism of increased spike train variability is not clarified in the present study. However, theoretical studies have shown that inputs to the GPe/GPi shift spike timings to desynchronize neural activity³⁷ and that the change of synaptic strength between the STN and GPe introduces noise through chaotic dynamics in the STN-GPe network³⁸. (p. 17, lines 307-311)

8. “Loss of excitatory inputs from the STN” (line 297) seems an overstatement. Perhaps more appropriate to say reduction in excitatory inputs.

We modified the phrase in Discussion on line 318-319:

Reduction in excitatory inputs from the STN to the GPe dramatically enhanced pauses (Fig. 4f), ... (p. 17, lines 320-321)

9. Lines 304-305: replace “65-75%” for “25-35%”, to reflect the proportion of reduction observed.

Based on the reviewer’s recommendation, we modified the phrase in Discussion:

... because reduction of STN activity was moderate (reduced neuronal activity by

25-35%) and ... (p. 19, lines 351-352)

10. The discussion around mechanisms of action of DBS (lines 343-346) is slightly out of context for this paper and, also, outdated. There is more recent evidence implicating many mechanisms that could account for the effects of STN-DBS. Consider updating (or perhaps removing) this part.

We toned down the description on STN-DBS in Discussion:

Diminished transmission via the *direct* pathway and enhanced transmission via the *indirect* pathway lose movement-related inhibition in the GPi^{5,52-54}, resulting in akinesia; however, the baseline firing rate in the GPi does not necessarily increase^{9,53-55}. The firing pattern also changes dramatically in PD; spike trains of many STN/GPe/GPi neurons exhibit correlated, oscillatory activity at the β frequency^{52,56}. Both lesion/suppression and DBS of the STN exhibit therapeutic effects on PD symptoms^{15-17,53,57}; these procedures may antagonize these GPe/GPi activity changes observed in PD. Chemogenetic manipulation in the present study moderately suppresses the STN activity and would also have beneficial effects on PD symptoms. (p. 20, lines 370-377)

11. Legend to fig 6: "Ratios of STN/GPe/Gpi neurosn exhibiting significant excitation or inhibition during involuntary movements..." Since the modulation occurs before the onset of the involuntary movement, perhaps may be better to say "Ratios of STN/GPe?Gpi neurons exhibiting significant excitation or inhibition before the occurrence of involuntary movements

Following the reviewer's recommendation, we modified the phrase in legend of Figure 6:

Ratios of STN/GPe/GPi neurons exhibiting significant excitation or inhibition before the occurrence of involuntary movements ... (p. 53, lines 999-1000)

Refs cited:

1 Yu, X., Nagai, J. & Khakh, B. S. Improved tools to study astrocytes. *Nature Reviews Neuroscience*, doi:10.1038/s41583-020-0264-8 (2020).

2 Upright, N. A. & Baxter, M. G. Effect of chemogenetic actuator drugs on prefrontal cortex-dependent working memory in nonhuman primates. *Neuropsychopharmacology*, doi:10.1038/s41386-020-0660-9 (2020).

3 Deffains, M. et al. In vivo electrophysiological validation of DREADD-based modulation of pallidal neurons in the non-human primate. *Eur J Neurosci*,

doi:10.1111/ejn.14746 (2020).

4 Nagai, Y. et al. Deschloroclozapine, a potent and selective chemogenetic actuator enables rapid neuronal and behavioral modulations in mice and monkeys. *Nat Neurosci* 23, 1157-1167, doi:10.1038/s41593-020-0661-3 (2020).

5 Eldridge, M. A. et al. Chemogenetic disconnection of monkey orbitofrontal and rhinal cortex reversibly disrupts reward value. *Nat Neurosci* 19, 37-39, doi:10.1038/nn.4192 (2016).

6 Raper, J. et al. Metabolism and Distribution of Clozapine-N-oxide: Implications for Nonhuman Primate Chemogenetics. *ACS Chem Neurosci*, doi:10.1021/acscchemneuro.7b00079 (2017).

7 Wilson, C. J. Active decorrelation in the basal ganglia. *Neuroscience* 250, 467-482, doi:10.1016/j.neuroscience.2013.07.032 (2013).

12.

Reviewer #3 (Remarks to the Author):

The manuscript “Subthalamic nucleus stabilizes movements by reducing neural spike variability in monkey basal ganglia: chemogenetic study” by Hasegawa and colleagues presents the behavioral and neurophysiological changes occurring following the injection of chemogenetic ligands to the motor part of the STN. This method was used for a partial suppression of STN activity and led to changes in neuronal firing and to abnormal movements which appear in-line with the previously shown hemiballism which resulted from complete suppression using methods such as lesions. The study is very thorough and utilizes a method which is rarely implemented in monkeys demonstrating their great potential for exploring movement disorders.

Major comments:

1. The authors make a bold statement regarding the causal relation of the spike train variability and the movement stabilization. The data that they provide does not support this claim but rather demonstrates only a correlation between the two phenomena. It is unclear whether such a causal claim may even be potentially supported given the available tools and paradigms.

As the reviewer pointed out, our observation was correlational; with reduction of STN activity, high spike variability and unstable movements occurred together. However, the following rationales support their causal relationship. First, since the GPi is the output

nucleus of the basal ganglia, behavioral effects are presumably caused by the change in neural activity of GPi neurons. The baseline firing rate and movement-related activity in the GPi were only weakly affected (Fig. 3d-f), while the spike variability significantly increased in the GPi (Fig. 4b). Thus, we hypothesized that high spike variability in the GPi induced unstable movements. Second, the Fano factor of the GPi was significantly different between short- and long-MT trials, and the onset timings of the significant difference well preceded Movement onset in the POST period (Fig. 5). Regarding to this point, we modified the discussion as follows:

In the GPi, output nucleus of the BG, the baseline firing rate and movement-related activity were only weakly affected after reduction of STN activity (Fig. 3), while the spike variability significantly increased (Fig. 4). Moreover, our across-trial analysis showed that long MT was associated with high spike variability before Movement onset in STN/GPe/GPi neurons after reduction of STN activity, but no such trend was observed in terms of firing rate (Fig. 5). In long-MT trials, high spike variability in the STN preceded that in the GPe/GPi by 70-90 ms during the POST period (Fig. 5d). These observations suggest a causal relationship between reduction of STN activity, increased spike variability in the GPe/GPi, and unstable movements. Based on these results, we would like to propose a novel perspective on the STN function; Excitatory STN inputs stabilize spike timing in GPe/GPi neurons, reduce trial-to-trial variability during movements, and contribute to performance of rapid and stable movements. Reduction of such excitatory inputs increases spike variability in GPe/GPi neurons and disturbs movements (Supplementary Fig. 14). (p. 18, lines 335-347)

2. The monkeys display a very different motor state following the chemogenetic manipulation which manifests by different movements which are partially recorded. It is not clear that the variability the GPe/GPi activity is not an epiphenomenon of the changes in neuronal activity due to the movements themselves throughout the session.

For movement related activity in the ET and SI trials, causal relationship between the variability the GPe/GPi activity and the movement changes was discussed above.

For neuronal activity related to involuntary movements, we think that the spike variability of the GPe/GPi is not an epiphenomenon of movements because firing rate modulation preceded the onset of involuntary movements in most of GPe/GPi neurons:

Neural activity in the STN/GPe/GPi exhibited similar modulation (i.e., increase or decrease) between in involuntary movements and during the task, and such

modulation well preceded the onset of involuntary movements. These results suggest that involuntary movements are induced via the following neural mechanism: 1) In the normal state, excitatory inputs from the STN stabilize GPe/GPi activity; 2) Reduction of excitatory inputs from the STN increases pauses and spike variability in the GPe/GPi; and 3) Coincident pauses/spikes across GPi neurons could induce disinhibition in the thalamocortical activity and cause immature movements, resulting in involuntary movements (Supplementary Fig. 14). (p. 19, lines 353-361)

3. There are multiple previous studies of the effect of lesions or chemical blocking of the STN and the formation of abnormal movements. It would be useful to explicitly compare (potentially also quantitatively) their behavioral and neurophysiological results in a table/graph format with the findings of the current study to emphasize the novel findings.

We added Supplementary Table 1 to summarize the experiments involving lesions or inactivation of the STN in normal sub-human primates.

4. The assumption presented in the discussion that the variability is derived from network transitions is speculative as it does not seem to be supported by the results.

We removed the part of Discussion the reviewer pointed out. Instead, we cited theoretical studies that propose possible neural mechanisms to induce spike train variability in the GPe and/or GPi in Discussion:

Reduction of STN activity increased spike train variability in STN/GPe/GPi neurons (Fig. 4). The mechanism of increased spike train variability is not clarified in the present study. However, theoretical studies have shown that inputs to the GPe/GPi shift spike timings to desynchronize neural activity³⁷ and that the change of synaptic strength between the STN and GPe introduces noise through chaotic dynamics in the STN-GPe network³⁸. (p. 17, lines 307-311)

Minor comments:

1. Changes in the expression of oscillations, i.e. spectral analyses are not presented.

We added the spectrum analyses for the STN/GPe/GPi in Fig. 4c.

We also examined oscillatory activity changes by the power spectrum density function averaged across neurons after normalization with a local shuffling method (see Methods for details), and observed no oscillatory activity changes

with reduction of STN activity (Fig. 4c). (pp. 13-14, lines 250-252)

2. The relation of the presented results to isolated E-I networks is unclear as in the presented case the variability is highly affected by the “external noise” and not by the specific GPe-STN balance.

As a possible mechanism to induce spike train variability, we added the following statement in Discussion:

Possibly, spike train variability is increased by higher sensitivity to external noise.
(p. 17, lines 311-312)

3. A few minor errors:

Page 6, Line 88: contralateral  ipsilateral

Page 15, line 273: marginally significant  not significant

We corrected all phrases the reviewer pointed out.

REVIEWER COMMENTS

Reviewer #1 (Remarks to the Author):

The authors have highly improved their manuscript and included several new, important control analyses. I have a few remaining points.

Related to Major 1: It is very nice to see the time course of firing rate and pattern changes in Supplementary Fig. 11. To get some insight into what variable may be related to the motor effects, could the authors also add the time course of motor performance (success rate, reaction time, movement time, occurrence of involuntary movements) to this figure? It would be interesting to see whether motor behavior dropped throughout the whole time course (similarly to STN rate changes) or rather soon after DCZ administration (similarly to pattern changes).

Related to Major 2: The new Supplementary Fig. 2 is very helpful to see in which regions cells were transduced, and it is great to see a clear maximum in STN. However, given that cells were also transduced in thalamus, in particular in the lateral nuclei, couldn't motor behavior be mediated via direct effects on thalamic spiking? This should be discussed in the manuscript, especially in the light of very subtle changes in GPi activity.

Related to Major 4: Supplementary Fig. 14 does not add to this discussion. The critical point (how do firing pattern changes in GPi translate into movement?) is represented only as a yellow arrow. I'd suggest to either remove this figure or to specify the process indicated by the yellow arrows.

Furthermore, as an alternative to the thalamic pathway, could motor behavior here also be influenced via the brainstem?

My remaining concerns have been fully addressed.

Reviewer #2 (Remarks to the Author):

1. In the rebuttal letter, in response to the concern about the description of the transgene expression, the authors state that "However, since the inhibitory DREADD receptor, hM4Di, was presumably cleaved from GFP by 2A self-cleaving peptide, hM4Di could be differently localized within a cell." This is an

understandable limitation of the study, and this statement (or similar) should be included in the manuscript.

2. The images in figure 2f show GFP+ processes that are likely axonal terminals from STN axons. However, lacking additional confirmation that these are, indeed, axonal terminals, the description should be more cautious and use the adjective 'putative' when referring to these terminals throughout the manuscript.

3. In supplementary figure 2c: It is unclear how the data in the right part of the figure was obtained. The legend states "ratios of EGFP-positive cells among all NeuN-positive cells, " Does this refer to all cells in each of the quadrants shown in the figures? How many sections were analyzed for the data shown in this figure?

4. The revised version of the manuscript, includes a paragraph regarding the timing of the electrophysiological recordings (page 19) as follows: "The concentration of DCZ in the cerebrospinal fluid is maximum at 90 min after administration³³, and a clozapine, a presumed active ligand converted from CNO51, would have the highest concentration later." Please specify what would be the 'later' time. The authors can refer to the information provided in published work (for example, Raper et al 2017).

5. Clarify what is "POST" (right-most data point) in figure 1d. I assume this is the average of the recording times after the systemic injections, this should be indicated in the figure legend.

6. While reading the revised version, I was wondering if the authors would like to comment on how their results relate to the observations made in the paper by Elias et al ("Balance of Increases and Decreases in Firing Rate of the Spontaneous Activity of Basal Ganglia High-Frequency Discharge Neurons", 2008). It may be appropriate to include this paper in the Discussion of results.

7. A few remaining minor points:

a. Please indicate how the "High-frequency" and "low-frequency" discharge neurons defined. In other words, what were the ranges of firing and bursting to classify a cell in each category?

b. Line 184, the section heading reads "GPe/GPi movement-related activity was weakly affected" I suggest adding "...after administration of chemogenetic ligands"

c. Similarly, heading in line 227 could read "Spike trains variability..... after administration of chemogenetic ligands"

Reviewer #3 (Remarks to the Author):

The revised manuscript "Subthalamic nucleus stabilizes movements by reducing neural spike variability in monkey basal ganglia: chemogenetic study" by Hasegawa and colleagues includes a large number of changes and analyses of the data which fully address the issues raised regarding the previous version.

The addition of the supplementary figures and table create a more comprehensive presentation of the key concepts presented in the manuscript.

Reviewer #1 (Remarks to the Author):

The authors have highly improved their manuscript and included several new, important control analyses. I have a few remaining points.

Related to Major 1: It is very nice to see the time course of firing rate and pattern changes in Supplementary Fig. 11. To get some insight into what variable may be related to the motor effects, could the authors also add the time course of motor performance (success rate, reaction time, movement time, occurrence of involuntary movements) to this figure? It would be interesting to see whether motor behavior dropped throughout the whole time course (similarly to STN rate changes) or rather soon after DCZ administration (similarly to pattern changes).

We analyzed the time course of behavioral changes and added Supplementary Fig. 11b. The behavioral changes occurred rather later than the firing pattern changes as described in Results:

Behavioral changes tended to occur after neural activity changes (2.9-9 min after DCZ administration; Supplementary Fig. 11b), suggesting that the high spike train variability was not due to behavioral changes. (p. 14, lines 253-255)

Related to Major 2: The new Supplementary Fig. 2 is very helpful to see in which regions cells were transduced, and it is great to see a clear maximum in STN. However, given that cells were also transduced in thalamus, in particular in the lateral nuclei, couldn't motor behavior be mediated via direct effects on thalamic spiking? This should be discussed in the manuscript, especially in the light of very subtle changes in GPi activity.

We added the discussion regarding to transduced cells in the thalamus in the figure legend of Supplementary Figure 2:

A small fraction of transduced cells were found dorsal to the STN such as the lateral nuclei of the thalamus (Lat. n.), thalamic reticular nucleus (Rt), and zona incerta (ZI). Transduced cells in the thalamus could have negligible effects because they avoided the forelimb region of the thalamus^{1,2} and because thalamic cooling did not induce involuntary movements³. (figure legend of Supplementary Fig. 2)

Related to Major 4: Supplementary Fig. 14 does not add to this discussion. The critical point (how do firing pattern changes in GPi translate into movement?) is represented

only as a yellow arrow. I'd suggest to either remove this figure or to specify the process indicated by the yellow arrows.

Furthermore, as an alternative to the thalamic pathway, could motor behavior here also be influenced via the brainstem?

Based on the reviewer's suggestion, we elaborated our model in the figure legend of Supplementary Fig. 14.

Supplementary Fig. 14 | Hypothetical role of the STN on motor control. a, Neural activity of the BG in the normal state. The STN interacts with the GPi to generate stable output in a resting state and coordinated firing rate changes during movements. Decreased firing rates in the GPi disinhibit the thalamocortical/brainstem motor center and release appropriate movements at the appropriate timing, while increased firing rates inhibit the motor center and suppress unnecessary movements, leading to stable movements. **b,** Neural activity during the reduction of STN activity. Spike trains of GPi neurons become variable during movements, leading to unstable movements. Spontaneous fluctuation of spike trains of GPi neurons, especially coincident pauses across GPi neurons, in a resting state could disinhibit the thalamocortical/brainstem motor center and cause immature movements, resulting in involuntary movements. (figure legend of Supplementary Fig. 14)

Also, we modified the statement in Discussion:

Reduction of such excitatory inputs increases spike train variability in GPe/GPi neurons, and unstable GPi activity changes during movements disturb movements. (p. 19, lines 354-355)

Coincident pauses across GPi neurons could induce disinhibition in the thalamocortical/brainstem motor center and cause immature movements, resulting in involuntary movements. (p.19, lines 367-369)

My remaining concerns have been fully addressed.

Reviewer #2 (Remarks to the Author):

1. In the rebuttal letter, in response to the concern about the description of the transgene expression, the authors state that "However, since the inhibitory DREADD receptor,

hM4Di, was presumably cleaved from GFP by 2A self-cleaving peptide, hM4Di could be differently localized within a cell.” This is an understandable limitation of the study, and this statement (or similar) should be included in the manuscript.

Based on the reviewer’s suggestion, we added the statement in the figure legend of Supplementary Fig. 2:

...and putative axonal terminals in the GPe/GPi (f, arrows). However, hM4Di could be localized differently from EGFP within a cell, since hM4Di was presumably cleaved from EGFP by 2A self-cleaving peptide. (figure legend of Supplementary Fig. 2)

2. The images in figure 2f show GFP+ processes that are likely axonal terminals from STN axons. However, lacking additional confirmation that these are, indeed, axonal terminals, the description should be more cautious and use the adjective ‘putative’ when referring to these terminals throughout the manuscript.

We fixed the manuscript according to the reviewer’s suggestion.

3. In supplementary figure 2c: It is unclear how the data in the right part of the figure was obtained. The legend states “ratios of EGFP-positive cells among all NeuN-positive cells, “ Does this refer to all cells in each of the quadrants shown in the figures? How many sections were analyzed for the data shown in this figure?

We added the detailed description in the figure legend of Supplementary Figure 1:

Frontal sections were divided into 0.5 mm × 0.5 mm squares along the dorsal-ventral (DV) and medial-lateral (ML) axes. Numbers of transduced neurons and NeuN-positive cells were counted, and transduction efficiencies, ratios of transduced neurons among all NeuN-positive cells, were calculated at each square. Transduction efficiencies were averaged along the anterior-posterior axis separately in anterior (3 sections) and posterior (4 section) STN regions and plotted along the DV and ML axes (right). (figure legend of Supplementary Fig. 1)

4. The revised version of the manuscript, includes a paragraph regarding the timing of the electrophysiological recordings (page 19) as follows: “The concentration of DCZ in the cerebrospinal fluid is maximum at 90 min after administration³³, and a clozapine, a presumed active ligand converted from CNO51, would have the highest concentration later.” Please specify what would be the ‘later’ time. The authors can refer to the information provided in published work (for example, Raper et al 2017).

We modified the statement in Discussion:

The concentration of DCZ in the cerebrospinal fluid is maximum at 90 min after administration³³, and clozapine, a presumed active ligand converted from CNO⁵², would have higher concentration later (>90 min after administration)⁵³. (p. 19, lines 370-372)

5. Clarify what is “POST” (right-most data point) in figure 1d. I assume this is the average of the recording times after the systemic injections, this should be indicated in the figure legend.

We clarified the definition of POST in the figure legend of Figure 1:

STN activity was normalized based on activity during the PRE period (from –15 to 0 min) and averaged in 5-min bins (left) and in the POST periods (right; from 15 to 25 min after CNO administration; from 10 to 25 min after DCZ/VEH administration).

6. While reading the revised version, I was wondering if the authors would like to comment on how their results relate to the observations made in the paper by Elias et al (“Balance of Increases and Decreases in Firing Rate of the Spontaneous Activity of Basal Ganglia High-Frequency Discharge Neurons”, 2008). It may be appropriate to include this paper in the Discussion of results.

We added the following statement in Discussion:

GPe/GPi neurons lacked phasic firing rate increases at rest without phasic excitatory STN inputs³⁷, suggesting that GPe/GPi firing patterns are sensitive to STN inputs. (p. 17, lines 314-316)

7. A few remaining minor points:

a. Please indicate how the “High-frequency” and “low-frequency” discharge neurons defined. In other words, what were the ranges of firing and bursting to classify a cell in each category?

We added the definitions for each neuron type in Table 1:

Based on firing rates and patterns, GPe and GPi neurons were classified as ‘high-frequency discharge and pause’ (HFD-P) GPe (mean firing rate ≥ 20 Hz), ‘low-frequency discharge and burst’ (LFD-B) GPe (< 20 Hz), ‘high-frequency discharge’ (HFD) GPi (≥ 20 Hz), or ‘low-frequency discharge’ GPi neurons (< 20 Hz). (legend of Table 1)

b. Line 184, the section heading reads “GPe/GPi movement-related activity was weakly affected” I suggest adding “...after administration of chemogenetic ligands”

We modified the subheading under the restriction of the number of characters that can be used for each subheading (≤ 60 characters):

GPe/GPi movement-related activity was weakly affected after CNO/DCZ (p. 10, line 185)

c. Similarly, heading in line 227 could read “Spike trains variability..... after administration of chemogenetic ligands”

We modified the subheading using only ≤ 60 characters:

STN/GPe/GPi spike train variability increased after CNO/DCZ (p. 12, line 229)

Reviewer #3 (Remarks to the Author):

The revised manuscript “Subthalamic nucleus stabilizes movements by reducing neural spike variability in monkey basal ganglia: chemogenetic study” by Hasegawa and colleagues includes a large number of changes and analyses of the data which fully address the issues raised regarding the previous version. The addition of the supplementary figures and table create a more comprehensive presentation of the key concepts presented in the manuscript.

REVIEWER COMMENTS

Reviewer #1 (Remarks to the Author):

I am afraid the last revision, in particular the time courses of behavioral data which are now shown in Supplementary Fig. 11, has raised new issues related to my previous concerns.

Related to Major 1: The authors now write “Behavioral changes tended to occur after neural activity changes (2.9-9 min after DCZ administration; Supplementary Fig. 11b), suggesting that the high spike train variability was not due to behavioral changes. (p. 14, lines 253-255)”. Isn’t the authors’ hypothesis that a reduction in STN firing rate and an increase in GPi spiking variability cause behavioral changes? How can these different time courses of neural activity changes and changes in behavior then be explained?

Related to Major 2: Did the authors show evidence that in the present study there were no transduced cells in the forelimb region of the thalamus? Otherwise, the alternative explanation of direct effects on thalamus should explicitly be mentioned in the manuscript, not only in the supplement. Effects on thalamic spiking may not necessarily be negligible as currently indicated by the authors.

Related to Major 4: I am skeptical about the mechanism suggested in Supplementary Fig. 14 for the following reasons. (1) If reduced STN activity causes pattern changes in GPi and thereby changes in behavior, why are the time courses for STN firing rate decrease, GPi pattern change, and behavioral change (Supplementary Fig. 2) not similar? This relates also to Major 1. (2) Changes in GPi activity after DCZ administration are very low, also in contrast to changes in neural activity in STN and GPe. Alternative to the pathway suggested by the authors, midbrain structures targeted by basal ganglia nuclei (for example, PPN receiving STN inputs) could bypass GPi. If the authors hypothesize that coincident pauses in GPi drive the behavioral changes, they should test for a correlation of coincident GPi pauses with behavioral changes. As far as I can see, coincident pauses in GPi have not been investigated.

Reviewer #1 (Remarks to the Author):

I am afraid the last revision, in particular the time courses of behavioral data which are now shown in Supplementary Fig. 11, has raised new issues related to my previous concerns.

Related to Major 1: The authors now write “Behavioral changes tended to occur after neural activity changes (2.9-9 min after DCZ administration; Supplementary Fig. 11b), suggesting that the high spike train variability was not due to behavioral changes. (p. 14, lines 253-255)”. Isn’t the authors’ hypothesis that a reduction in STN firing rate and an increase in GPi spiking variability cause behavioral changes? How can these different time courses of neural activity changes and changes in behavior then be explained?

We think that activity of a certain number of neurons must be changed before behaviors were changed. Since it took a few minutes before the population neural activity reached a plateau from the onset (Supplementary Fig. 11a; e.g., Fano factor of GPe neurons), DCZ would affect neural activity gradually over the time course of several minutes. To clarify this point, we modified the statement in the Results:

Behavioral changes tended to occur after neural activity changes (2.9-9.0 min after DCZ administration; Supplementary Fig. 11b), suggesting that the high spike train variability entrained a group of neurons, and then behaviors changed with time lag. (p. 14, lines 253-255)

Related to Major 2: Did the authors show evidence that in the present study there were no transduced cells in the forelimb region of the thalamus? Otherwise, the alternative explanation of direct effects on thalamus should explicitly be mentioned in the manuscript, not only in the supplement. Effects on thalamic spiking may not necessarily be negligible as currently indicated by the authors.

We added Supplementary Figure 2h to show the distribution of transduced cell in the lateral nuclei of the thalamus. We modified the statement in the legend of Supplementary Figure 2:

h, Distribution of transduced cells within the Lat. n. Transduced cells in the densest section were plotted, and those outside the Lat. n. were omitted for clarity. They were found in the most ventrolateral part of the Lat. n. and avoided its central part presumably representing the forelimb based on the somatotopic organization of the thalamus along its mediolateral axis¹⁻³. (legend of Supplementary Fig. 2)

Also, we included the following statement in the Results:

Transduced cells in the thalamus avoided its forelimb region (Supplementary Fig. 2h) and would have negligible effects on forelimb movements. (p. 5, lines 76-77)

Related to Major 4: I am skeptical about the mechanism suggested in Supplementary Fig. 14 for the following reasons. (1) If reduced STN activity causes pattern changes in GPi and thereby changes in behavior, why are the time courses for STN firing rate decrease, GPi pattern change, and behavioral change (Supplementary Fig. 2) not similar? This relates also to Major 1. (2) Changes in GPi activity after DCZ administration are very low, also in contrast to changes in neural activity in STN and GPe. Alternative to the pathway suggested by the authors, midbrain structures targeted by basal ganglia nuclei (for example, PPN receiving STN inputs) could bypass GPi. If the authors hypothesize that coincident pauses in GPi drive the behavioral changes, they should test for a correlation of coincident GPi pauses with behavioral changes. As far as I can see, coincident pauses in GPi have not been investigated.

We believe that the phasic activity change plays a role in unstable movements and involuntary movements because of the following reasons: 1) High spike train variability before Movement onset was correlated with unstable movements (Fig. 5). 2) Although we did not examine the coincidence of GPi pauses and behavioral changes, a subset of GPe/GPi neurons exhibited phasic activity increase/decrease preceding the occurrence of involuntary movements (Fig. 6). However, as the reviewer pointed out, our experiments cannot exclude the involvement of other pathways. Hence, we described the limitation of our studies and modified the description as follows in Discussion:

1) In the normal state, excitatory inputs from the STN stabilize GPe/GPi activity in the resting state and suppress involuntary movements; 2) Reduction of excitatory inputs from the STN increases spike train variability in the GPe/GPi; and 3) Coincident activity changes of GPe/GPi neurons, which are similar to those during voluntary movements, may occur and cause involuntary movements through the BG-thalamo-cortical pathway. However, the possible involvement of the projections from the BG, such as from the GPi and STN, to the brainstem motor centers could not be excluded. Further studies using pathway-specific manipulations of neural activity could test our hypothesis. (p. 19, lines 365-372)

Also, the description in the legend of Supplementary Figure 14 was modified:

a, Neural activity of the BG in the normal state. The STN interacts with the GPe/GPi to generate stable activity in the resting state and coordinated firing rate changes during movements (yellow), contributing to suppression of involuntary movements and stable movements, respectively. **b**, Neural activity during the reduction of STN activity. Spike trains of GPe/GPi neurons become variable. Increased spike train variability during movements (yellow) leads to unstable movements. In the resting state, increased spike train variability may increase the chance of coincident activity changes (green) similar to those during voluntary movements, resulting in involuntary movements. (legend of Supplementary Figure 14)